# The social organization of the Asian weaver ant colonies: A natural enemy novel sub-castes worker's functional activity findings

Moïse Pierre Exélis[1,2]*, Rosli Ramli[1]*, Roslinazairimah Zakaria[3], Azarae Hj Idris[1], Zubaidah Ya'cob[4], Rabha W. Ibrahim[5,6,7], Salmah Yaakop[8], Nurul Wahida Othman[8]

1 Institute of Biological Sciences, Faculty of Science, Universiti Malaya. Lembah Pantai, Kuala Lumpur, Malaysia, 2 Direction de l'Enseignement Supérieur et de la Recherche Espace Etudiants, Hôtel de la Collectivité Territoriale de la Martinique (CTM), Rue Gaston Defferre - Cluny - CS–, Fort-de-France Martinique, 3 Centre for Mathematical Sciences, Universiti Malaysia Pahang Al-Sultan Abdullah. Lebuh Persiaran Tun Khalil Yaacob, Kuantan Pahang Malaysia, 4 Higher Institution Centre of Excellence (HICoE) Tropical Infectious Diseases Research & Education Centre (TIDREC), Universiti Malaya. Lembah Pantai, Kuala Lumpur, Malaysia, 5 Department of Computer Science and Mathematics, Lebanese American University, 13, Beirut, Lebanon, 6 Department of Mathematics, Near East University, Mathematics Research Center, Near East Boulevard, PC, Nicosia/Mersin 10-Turkey, 7 Information and Communication Technology Research Group, Scientific Research Center, Al-Ayen University, Thi-Qar, Iraq, 8 Department of Biological Science and Biotechnology, Centre for Insect Systematics, Faculty of Science and Technology, Universiti Kebangsaan Malaysia, Bangi Selangor, Malaysia

* rosliramli@um.edu.my (RR); exelis.pierre@gmail.com (MPE)

## Abstract

An arboreal ant species by nature, the Asian weaver ant *Oecophylla smaragdina* F., (Hymenoptera: Formicidae) colony's social structure composition was investigated in depth. Brood and barrack nests were collected from the African oil palm (*Elaeis guineensis*) canopies and Limau kasturi (*Citrus microcarpa*) orchards, and dissected. All caste's morphological traits were examined stereo-microscopically. The workers' width and length measurements of the separately dissected head, thorax, frontal view, abdomen, and body full side view sizes were recorded. All colonies comprise a founding queen laying thousands of eggs stored in a protective yellowish, unknown sticky substance (shining reflection), with reproductive winged green and newly emerged yellow queens, adult drone males, and wingless workers along their immature pupae and larvae arranged in woven, solid silken chambers (brood nests exclusively). Besides the traditionally known caste of minor and major workers, five polymorphic individuals comprising two unidentified novel sub-castes of intermediate workers and one sub-caste of major workers were described. The full body and abdomen lengths are proposed as dominant predicting factors differentiating among the five sub-castes. The discovery of a multimodal size frequency distribution model contrasts with the classical archetypical bimodal systems in ants. Intermediate workers foraging outside the nest revealed reconnaissance autonomy and aggressive behaviors that aided larger workers in securing the territorial perimeter. Bigger

**Data availability statement:** All relevant data for this study are publicly available from the Mendeley Data repository (https://doi.org/10.17632/rh35my7tyt.1).

**Funding:** The author(s) received no specific funding for this work.;

**Competing interests:** The authors have declared that no competing interests exist.

workers occupied the first defensive layers of the colony's territorial frontier, while the intermediate workers maintained their stance at a closer nest distance. Major workers systematically acted as leaders-supervisors by removing individuals of smaller size during overheating exposure. Due to their short lifespan and segregated nests, it is difficult to collect adult males in wide plantations. A stable and average mature three-year-old colony produces several reproductive individuals monthly. The mean number of emerging queens is higher in older colonies (scarcity of males) and lower for younger colonies (queens-males averages are correlated). The queen production increases with higher rainfall and relative humidity. This study identified three novel worker sub-castes: one major intermediate, two intermediate in size. These findings contribute to a better understanding of the overall worker's functional activity. The Asian weaver ant demonstrates adaptive measures in response to challenging abiotic factors (temperature), defying classical labor division rules.

## 1. Introduction

The Asian weaver ant, *Oecophylla smaragdina* F., (Hymenoptera: Formicidae) is a dominant arboreal ant species inhabiting forests [1] and the African oil palm crops *Elaeis guineensis* Jacq. (Arecales: Arecaceae) canopies in Southeast Asia Malaysia [2–4]. Matured *Oecophylla smaragdina* colonies are polydomous (more than one nest per occupied host tree) and occupy multiple trees [5,6].

### 1.1 Colony's caste social structure composition

In ants, a caste is defined as a morphological distinction between individuals with specialized functions within the colony, thus referring to polymorphism [7]. Trible and Kronauer highlighted the exclusive significance of body size measurements for caste determination and distinction [8]. The mechanism underlying the unexplored concept of caste distinction remains poorly explored. Caste is defined as being associated with the effect of tissue growth variations during the pre-pupae and earlier pupae life cycle development stages, establishing the size-frequency range variation of individual ants within a species [8]. The colony caste structure composition of *Oecophylla smaragdina* is not completely documented [9], and further investigation may unveil more information on the workers' caste [10,11]). Muratore et al. adopted the definition of caste as appropriate in our current study (variety of individual female ants – workers and queens – determined by their distinctive morphological traits) [12]. Generally, among most ant species, inter-caste or sub-caste of individuals intermediate in size between the castes is scarce or absent [12]. Hence, we argue that caste-specific morphometric shape and size cannot represent a continuum of morphological variation but are the product of strongly conserved and genetically heritable traits [13]. A colony is defined by a founding depositing eggs, a dealate or ergatoid queen (shed wings), winged reproductive individuals composed of virgin queens, and drone

males. A matured colony that survived the founding stage develops into an average of half a million or more sterile individuals inhabiting several tree canopies with an extensive nesting network; these individuals include the adults, major and minor workers, and the immature stage of pupae, larvae, and eggs [5]. Sadly, only one study provides a comprehensive description of all castes of individuals collected from nests of the mango tree *Mangifera indica* L. (Sapindales: Anacardiaceae) and the pongame oil tree *Millettia pinnata* or *Pongamia pinnata* (L.) Pierre (Fabales: Fabaceae) in Malaysia [14]. Another report highlighted the extant third intermediate worker, whose average size is double that of minors and one-half that of majors [15]. The head width, length, thorax, body, and abdomen length were not evaluated. The colony composition, defined by a distinct caste system between reproductive and sterile individuals, requires additional consideration. The study of a multimodal size distribution of the worker caste in the *O. smaragdina* colony is novel. The Asian weaver ant is a valuable model system. We propose to verify the allometry changes subjected to a dimorphic or multimodal distribution of the worker caste (major-minor). We argue that the morphometric distinct worker features and their different related behavior (foraging, hunting, and predation) between novels identified intermediate size versus bigger size individuals are the prelude to justifying the existence of unreported sub-castes [12].

### 1.2  Labor division versus leadership function (sub-castes' behaviors)

In ants, the concept of labor division or polyethism is closely related to the individual's size class, defining for each caste a specific exclusive task ability function at the colony level to reach economic efficiency [16]. However, this concept is being questioned more as the term behavior flexibility for ants is becoming more appropriate now [17]. The traditional idea of any association between size and the suitability of task performance is denied for the more advanced acceptance of the environmental effects on ant's decision-making process and activity orientation and patterns [17]. Morphology matters ecologically in its framework, depending on periods of vulnerability linked to environmental parameters like temperature [18]. Ants are classified as social insects operating their daily activity without leadership [19]. Here we propose the hypothesis that such a concept might be more complex than it seems. We advocate the existence of a more advanced system of regulation in the Asian weaver ant, *O. smaragdina*, as a model. The supervision of major workers is reported during the foraging activity along the trails to monitor and control smaller-sized workers during the higher daily air temperature (Fig 4) [11]. Removing those intermediate workers was systematic, with major workers positioned at key positions, making them flexible to act whenever necessary to guarantee the safety of vulnerable colony individuals. The response of worker ants to temperature changes linked to their physiological traits resistance ability is reported as the maximum and minimum critical thermal limits ($CT_{max} - CT_{min}$) [20]. Certain ant species tend to forage in higher temperatures for a shorter activity period, and longer seasonal foraging acts are detected for ant species operating at lower temperatures.

### 1.3  Nest category – Reproductive cycle

A brood nest is characterized by the presence of all individual castes (reproductive: winged green queens, emerged callow yellow-winged future queens, drone males, and sterile major and minor workers, immature caste, i.e., pupae, larvae, and eggs) [14], typically in a high position in tree canopies [11]. A barrack nest contains only major workers in the lowest positions (height from the ground) at the tree canopy's periphery [21]. We speculated that thoroughly investigating the weaver ant progeny, brood, and barrack nest contents could reveal new information about the unreported worker sub-caste. We also predicted that such an evaluation would provide additional insight into how these novel individual workers sub-caste contributions influence colony-level activities that aid the territorial defense and survival of the colonies. As the size of adult emergence is dictated during the pupation growth development stage [22], we hypothesized that unreported existing sub-castes can be identified from collected dissected brood nests with preliminary visual discrimination of their morphological characteristics.

The knowledge of the Asian weaver ant mating mechanism process and reproductive individuals' emergence is at an infancy level and substantiated by only a single report [23]. Ant species mating phenology data in the function of seasonal shifts (rainfall interception and relative humidity annual fluctuations) are rare and valuable [24].

The objectives of the study were as follows: (1) to estimate the number of eggs laid per collected brood nest for each colony; (2) to examine the colony social structure composition by describing the average population size; (3) to describe the worker castes and identify the existence of distinct unreported major and intermediate workers sub-castes; (4) to evaluate the functional activity of the workers; and (5) to determine the yearly emergence pattern of the reproductive individual association to rainfall interception (RI mm/h) and relative humidity (RH %) fluctuations. Finally, the queen and male production was compared between younger (2–3 years old) and older colonies (above 3 years old).

## 2. Methodology and materials

### 2.1 Study sites

The main study was conducted at the oil palm plantation Felda Gunung Besout in Perak, Peninsular Malaysia (03°50'04"N 101°17'48" E) (Fig 1). Mango (*Mangifera indica*), key lime (Limau nipis, *Citrus aurantiifolia (Christm.)* Swingle (Sapindales: Rutaceae), lemon *Citrus limon* (L.) Osbeck (Sapindales: Rutaceae), Calamansi- Philippines lime (Limau kasturi; *Citrus microcarpa* Bunge), and durian *Durio zibethinus* L. (Malvales: Malvaceae) are among *Oecophylla* ants' favorite host mixed plants found on the estate within palm blocks [1]. The area experiences significant monthly precipitation fluctuations ranging from an average of 400 mm/month to more than 1000 mm/month. Additional sampling sites comprised oil palm estates located in Teluk Intan, Perak (Peninsular Malaysia) (3°49'07"N 100°58'59"E) and Felcra-Universiti Malaya plantations, Kota Tinggi Johor (Peninsular Malaysia) (2°02'03"N 103°51'50" E), Lahad Datu, Sabah (Borneo, East Malaysia) (5°01'52.4"N 118°25'33.2" E), and Saratok, Sarawak (Borneo, East Malaysia) (1°55'25.4"N 111°13'33.0" E) under the Malaysian Palm Oil Board (MPOB).

**2.2.1 Study species – Methods concept.** The Asian weaver ant, *O. smaragdina* (Fabricius, 1775), is a widely distributed species from the Solomon Islands (Pacific Ocean) to the subcontinent (South Asia), China (including the Himalayas Mountains), and Southeast Asia [25]. Their favorite habitats are dense tropical rainforests, oil palm plantations, and rich nectar-fruiting tree canopies, with surging populations occupying rural and urban areas [11].

Classification – Order: Hymenoptera. Family: Formicidae. Subfamily: Formicinae. Tribe: Oecophillini. Genus: *Oecophylla* Fabricius 1775. Type species: *Oecophylla smaragdina* Fabricius 1775; original designation by Fabricius (1775).

The concepts of the methods used in this study are followed closely by Trible and Kronauer [8,26] and Muratore et al [12], defining that size predicts caste with the hourglass concept. We propose modifications to Trible and Kronauer's

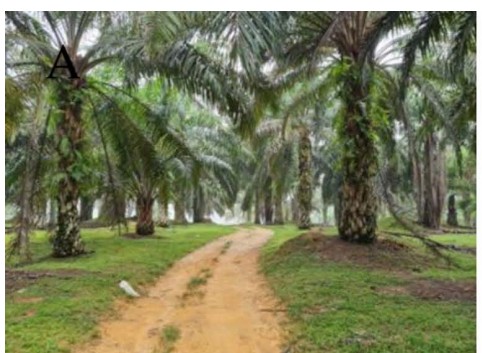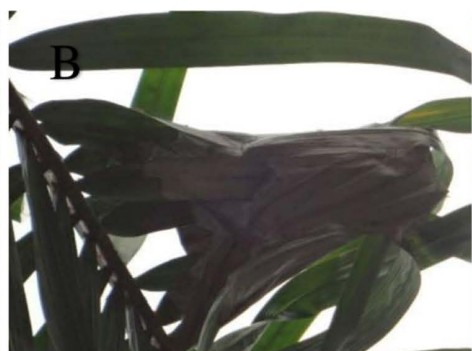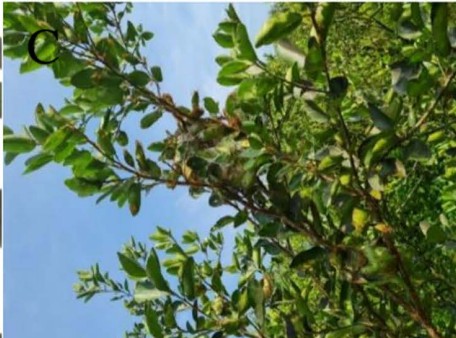

**Fig 1. Sampling site at Felda Gunung Besout Perak. (A)** The habitat of the study site is within oil palm blocks. **(B)** Brood nest collected from a palm frond in an elevated position. **(C)** A large brood nest was sampled and captured from a Calamansi tree orchard within a palm block.

method to refer to the caste differentiation mechanism following the multimodal distribution of the workers pupal body size stage, not the "late larval body size/mass" induction. Observations of 500 sampled pupae (100 for each identified class) proved that such conceptual orientation is founded on a positive match between each pupa and its relative adult form. In that sense, the existence of pupae exhibiting similarity to the widely overlapping size of novel intermediates and bigger size major reinforces this hypothesis, joining the findings of Anand et al [14], stating that caste traits modulation is genetically conserved, stable, and heritable but not an evolutionary vector. We adhere to caste differentiation as a function of size, as Trible and Kronauer [8] state. Muratore et al [12] used a multivariate Gaussian mixture model to determine the limits of worker size classes (sub-castes). In line with Trible and Kronauer's review [8], we propose the distinction of five inter-classes of individuals among the workers (datasets analysis findings – scaled from I to V) in the Asian weaver ant colony: minor (MW), intermediate 2 (IW2) and intermediate 3 (IW3), major intermediate worker (MIW) and soldier major big worker (MBW).

*O. smaragdina* forms a large colony of strong individual workers that colonize food crops by obeying the labor division concept. Traditionally, the workers' category consisted of the major and minor according to their body size. The labor tasks attribution is mainly foraging, hunting, territory permanent defense, and colony expansion by nest construction (major) [11]. Queens, brood care, and their inside nest maintenance are the sole responsibility of the smaller-sized minor.

**2.2.2 Nest collection and caste structure laboratory assessment.** Due to the difficulty of capturing a colony's egg-laying queen and the absence of male alates during sampling, other collection records from sampling sites are included in this evaluation. This is due to the possibility that the successful capture of dealate queens is only feasible through invasive, destructive methods, resulting in the extinction of the colony [5]. Previous studies conducted a five-kilometre-radius survey to detect the presence of mature *O. smaragdina* colonies [3,4]. Upon confirming the presence of large colonies exhibiting polydomous nesting behaviours in several palm trees within the plantations, a 1000-meter-wide transect was established along a path within the plantation. A minimum distance of 200 meters was maintained between palm tree canopies sampled to prevent the capture of nests from the same *O. smaragdina* colony. The antagonism test was conducted to detect distinct colonies. Between released and resident ants, ants from the same colony will pass, forage, and spontaneously communicate. Different colonies of ants will result in a conflict [27]. Three distinct palm tree-dwelling colonies, Felda I, Felda II, and UM III, were identified and tagged during the day. After sustained daytime rainfalls, three brood nests were collected from each colony during the late night to midnight period, 2230–0030 hours, corresponding to periods of low main worker activity determined in a separate study [11]. Felda I colony represented a large colony established on Calamansi (Limau kasturi; *Citrus microcarpa*) short trees of 2 m in height, forming an orchard with an average 100 m x 200 m land area within palm tree blocks. It was only feasible to collect nests during the daytime on short palms (3 m) using pyrethroids. Only samples from Felda I and UM III were reported (samples from Felda II suffered from misplacement during the COVID-19 pandemic, and many samples were dehydrated). Brood nests were identified based on a visual discrimination height parameter established in another study, located at an average position in palm tree canopies between 6 and 10 meters above the ground. A long hook was used to bring down the tall fronds slowly to cut the brood nests with a sharp cutter from the base of leaflets affixed to the rachis of each palm frond, and all weaver ants were collected in a rigid plastic bag (1 m x 60 cm). Pyrethrin knockdown was used to sample weaver ants to avoid ferocious stings and minimize individual escape and loss. Nests of *O. smaragdina* ants were dissected and classified according to the following castes after 24 hours of exposure to open air (protected from sun heat under a corridor roof – hence the strong disturbing smell of formic acid mix with pyrethrin disappeared): dealate queens, egg clusters, larvae, pupae, female alate virgin queens, male alate, major, intermediate, and minor workers [14]. The average position of barrack nests dissected with a keen surgical blade was determined to be between 3 and 4 meters below the lowest palm fronds, and their caste composition was recorded. Brood and barrack nests (two and one per colony, respectively) were used to evaluate the composition of the individual populations of three colonies Felda I, Felda II, and UM III, which helped determine the expected worker sub-castes (major, intermediate size, and minor workers). In the field, all varieties

of specimens that had not been exposed to the sun were separated, preserved in 95% ethanol, and transferred in 70% ethyl alcohol. The samples were transported to the Entomology Laboratory, Universiti Malaya, and the Centre for Insect Systematics, Universiti Kebangsaan Malaysia, for the morphometric measurements data collection (Fig 2A–2D).

**2.2.3 Estimation of egg number per nest.** In the first experiment, 20 nests representing ten colonies (2 nests per colony) were selected, with two primary brood nests per colony containing egg clusters encased in white silken chambers separated from other individual castes within the nest. Egg clusters must be collected carefully, as they are fragile (breaking or exploding under pressure) and attach to the interior nest wall. After filtration and drying of all specimens, the total number of eggs per nest was estimated by clusters (Fig 3A,B) using dissecting surgical needles and narrow forceps to separate eggs under a solution of ethyl alcohol 80% in a petri dish. Using thin needles and a 70% alcohol solution, the eggs poured into a tiny glass petri dish could be easily separated (Fig 4A). The separated embryos were then sucked directly into a microscope using a pipette. When it was challenging to separate eggs from each cluster due to their fragility, they were placed on white filter paper and counted visually with a magnifying device before being examined under a Nikon e200 compound microscope. The eggs were tallied using a stereomicroscope Nikon SMZ800N on a PC HP Compaq running Windows 7 Professional, connected to a Nikon digital sight DS-fi2 camera and NIS Elements version 4.0 imaging software. A ZEISS Stereo Discovery V20 microscope equipped with AxioCam MRc software Image Analyzer was also utilized.

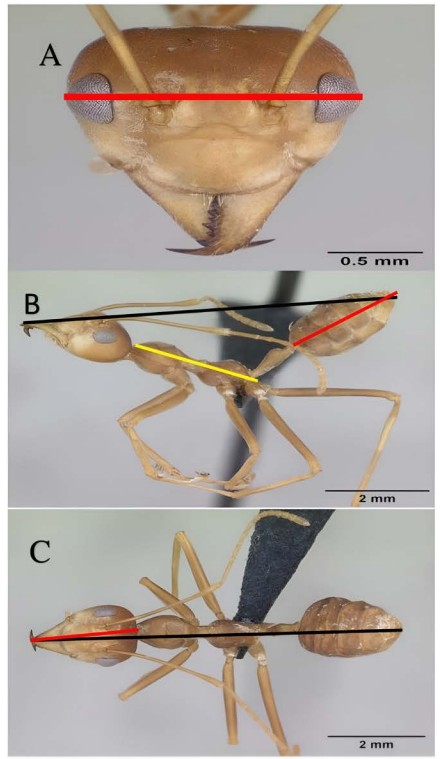
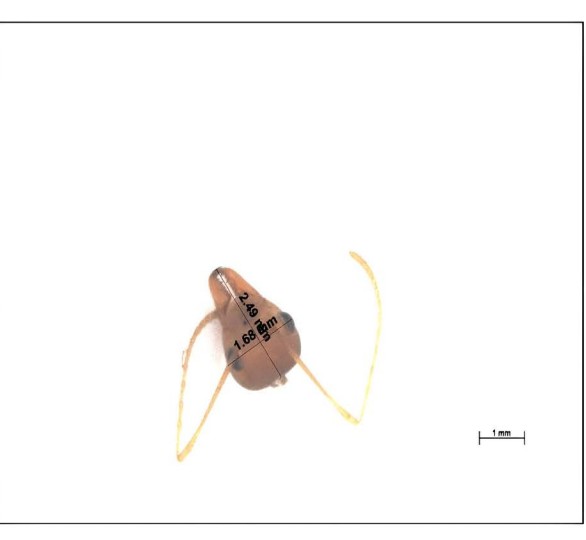

**Fig 2. Comparison between specimens.** A major worker specimen from Thailand was taken as a reference model (CASENT0173647, April Nobile) and matched with sampled individuals collected in Felda Gunung Besout plantations during this research. The measurements follow the three views shown: a frontal view (HW, red line – **A)**, a side view (TL, yellow line, AL, red line, and BL, black line – **B)**, and a dorsal view **(C)**. (HL, red line – BL, black line – **C)**. As for other measured body parts, the head section was dissected to minimize margin error occurrences (thorax and abdomen – **D)**. Images are accessible at www.antweb.org. AntWeb. Version 8.64.2. California Academy of Science, online at https://www.antweb.org. Accessed 22 September 2021.

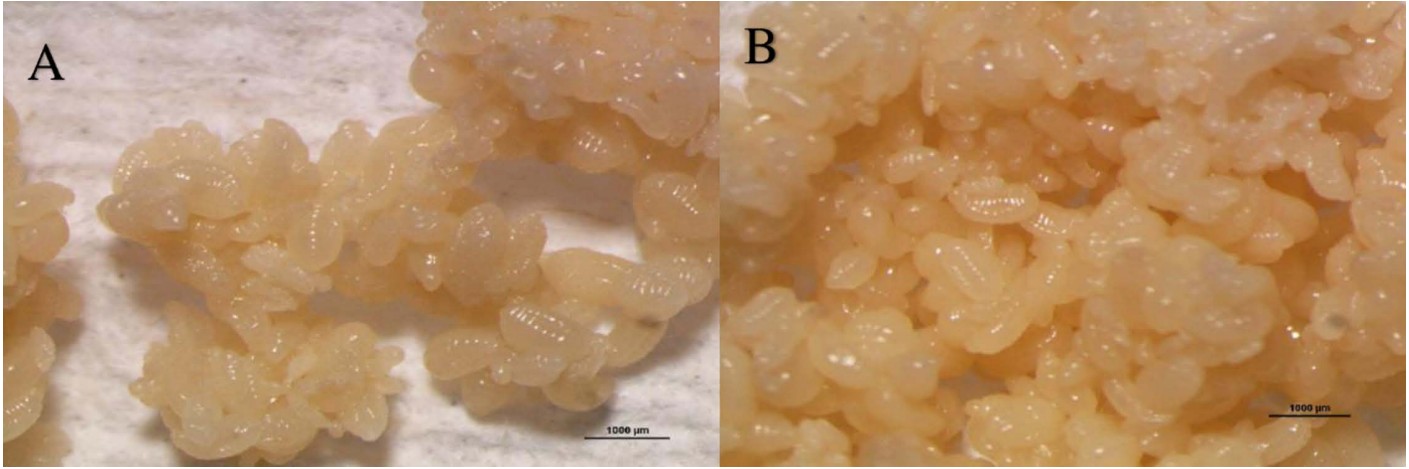

**Fig 3. *O. smaragdina* yellowish eggs clusters. (A)** Separated and tallied eggs from brood nests with a glossy appearance. **(B)** Compact egg clusters with a micron-sized standard. Examining using Nikon SMZ800N stereomicroscope. Photo credit: Exelis Moïse Pierre.

**2.2.4 Estimation of individual population size from matured colonies.** In the second experiment, we estimated the population from six nests (3 for each colony) from two sizable colonies inhabiting 8 and 12 palm trees, representing Felda I and UM III, respectively. The Felda I and UM III polydomous colonies comprised 33 and 74 geographically interconnected nests, respectively. The contents of every nest were once spread out on a white laboratory table. Each nest was meticulously dissected to obtain an accurate count of its inhabitants by directly counting all individual castes: adults (major, intermediate, and minor workers with winged reproductive of each sex), pupae (worker, male, and queen pupae), and larvae. By multiplying the mean of the total worker population for each colony by the total number of nests, the colony population size (PS) can be estimated on average: $PS = \sum x$ (Total nest number) x (TW), where TW represents the total worker values.

**2.2.5 Individual worker sub-caste measurements.** In the third experiment, the body and abdomen lengths (BL-AL) of the collected adult and pupae workers revealed distinct differences in body size. To determine the size distribution of workers within a colony, hundreds of individuals were chosen at random to have their body measurements recorded. The dehydrated specimens were fixed using a pin-holding stage. The dissected body parts were measured separately to avoid margin errors, giving the most accurate recorded values (Fig 2D). The sizes of the maximum distance across the eyes, head width (HW, transversal red arrow in Fig 2A), the maximum distance across the widest head part, head length (HL, longitudinal red arrow in Fig 1C), thorax length (TL-yellow arrow in Fig 2B), abdomen length (AL-red arrow in Fig 2B), and body length (BL, transversal red arrow in Fig 2C) were recorded. Obtained data prevailed as the estimation of the worker size variations identified into a size class scale from I to V, consisting of the names mentioned in section 2.2.1 (Table 1).

The morphometric measurements were performed using a stereomicroscope Nikon SMZ800N on a PC HP Compaq running Windows 7 Professional, attached to an integrated Nikon digital sight DS-fi2 camera and imaging software NIS Elements 4.0. The morphological characteristics were compared to images of an *O. smaragdina* specimen from Chonburi Pattaya, Thailand, on AntWeb (CASENT0173647). Using the stereomicroscope Nikon SMZ800N and aided by the Zeiss stereo discovery V20, a dissection was performed to identify any imperfect sightings or concealed body parts and measure the five traits by 10x, 20x, or 40x power converted to millimeters for workers and microns for eggs by conversion factor calibration to the microscope. All individuals exhibiting damaged body parts and a modified abdomen shape (compressed) were discarded. The dataset of body size records was pre-processed and filtered to reorganize and categorize them into distinct sub-castes based on the similarity of the values of the five variables to visualize the size distribution. The distribution will be either normal, bimodal, or multimodal.

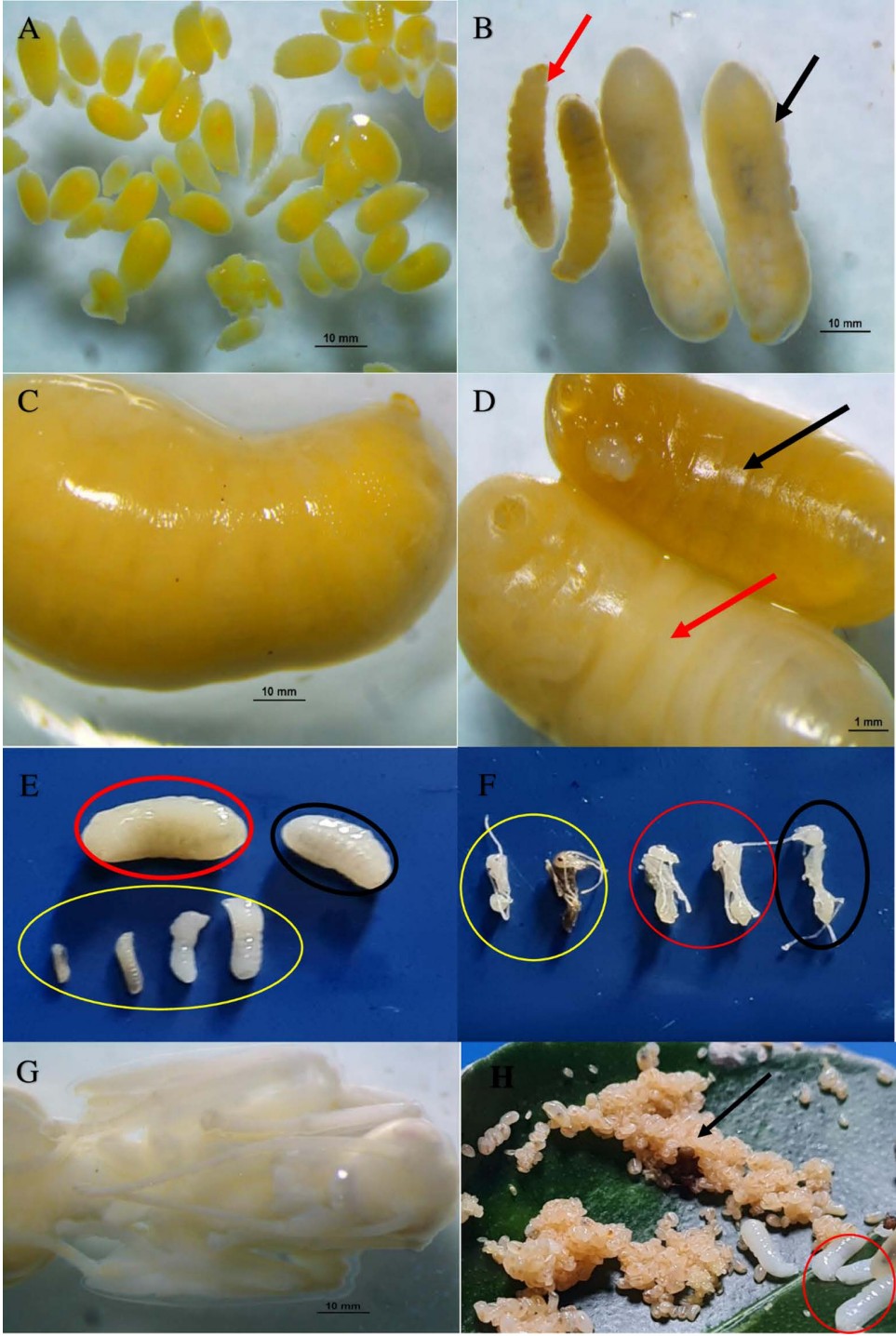

**Fig 4. Immature stage of *O. smaragdina* individuals.** *O. smaragdina* distinguished yellow eggs in black circles **(A)**. Major (black arrow) and subordinate worker larvae (red arrow) **(B)**. Big queen larvae **(C)**. Male drone larva (black circle) and female queen larva (red arrow) **(D)**. Queen (red circle) and male larvae (black circle). Major worker, intermediate 3, intermediate 2, and minor worker (yellow circle), from right to left, **(E)**. Workers' pupae with their distinctive sizes, from right to left: a major worker (black circle), an intermediate worker (red circle), and a minor worker (yellow circle). **(F)**. Queen pupae **(G)**. Identifies the distinction between eggs (black arrow) and larvae (red circle). **(H)**. All figures scales are 1 mm—examination via Nikon SMZ800N stereomicroscope (Photo credit: Exelis Moïse Pierre).

**Table 1. Worker morphological measurements – abbreviations.**

| Traits | Shorter name | Acronym |
|---|---|---|
| Maximum distance across eyes (Fig 2A, red line), frontal view | Eye to eye | HW |
| Head length (posterior cephalic margin to labrum), (Fig 2C, red line) | Head length | HL |
| Thorax length (mesosoma from the anterior prothorax margin to the insertion of the petiole) in side view (Fig 2B, yellow line) | Thorax length | TL |
| Abdomen length in side view (tip end to petiole insertion axe. Fig 2B, red line) | Abdomen length | AL |
| Body length in side view (abdomen to the mandible tip, Fig 2B, black line) | Body length | BL |

**2.2.6 Major – Intermediate workers evaluation – behaviors.** The specific function of the two identified intermediate and major intermediate workers sub-castes was studied using three mature, distinct colonies selected from separate palm tree blocks and designated colonies I, II, and III, which were monitored and evaluated (fourth experiment). The intermediate-size workers (an average of 10 per colony) were observed in the field on fronds, leaflets, trunks, and the ground to observe their foraging activity and distinguish their behavior from that of the major workers. A comparison was made with minor workers, typically restricted to the nest compound alone. The surveillance was conducted daily from 30 November to 6 December 2020 and 6–13 July 2022 during peak hours (1100–1300 hours & 1600–1800 hours) following Exélis et al [11]. Each observation lasted five minutes, and three observations were performed every hour (1100-1130-1155 hours). Photographs were captured to document the activities of workers outside nests. The major big worker (MBW) brought the smaller workers back to the canopy. At first sight, it looked like captured prey from a distinct colony. Still, after closer observation, we noticed that MBW is removing smaller-sized workers from the trunk and ground trails during the daily higher air temperature periods. Smaller workers were transported back to the nearest canopy nests. Consecutive monitoring during three days confirmed this behavior in the three distinct colonies.

**2.2.7 Monitoring reproductive caste emergence.** We evaluated the reproductive caste output throughout the year by selecting established colonies (fifth experiment). The plantation management confirmed that the occupied areas were chosen based on their known presence and that the average colony age was 3 years or older. From 18 to 24 months, these colonies start to generate adult sexual. In the sampling of reproductive individuals, such as winged virgin queens and male drones, from 2011 to 2022, two different methods were used: destructive sampling and direct surgical nest dissection. At the selected plantations described in section 2.1, the sampling of brood nests was typically done overnight between 2200 and 0100 hours. From 2011 to 2016, on a three-month basis, alternating between various months each year to cover the entire year. Nest sampling frequency decreased from 2018 to 2022 (November 2018, August 2019, January-March, and November-December 2020, January 2021, April-May-June to October 2022). We can acquire nests for every month of the year. The dataset displaying which month of the year had positive or negative emergence instances was created by recording and pooling the values. The Department of Irrigation and Drainage (Ministry of Environment and Water - https://publicinfobanjir.water.gov.my/hujan/?lang=en) provides the rainfall interception (RI) means (mm) dataset log and the relative humidity (RH %) recorded by a portable hygrometer digital recording device.

## 3. Statistical analysis

This section presents numerical and graphical statistical analyses, using 1% and 5% significance levels. The colony eggs production data followed a normal distribution, and the ANOVA test for comparing the means is suitable (Supporting information S1 Table). The Tukey Honest Significant Difference (HSD) test is a post hoc multiple comparison test used to determine which group means are significantly different after performing an ANOVA (Supporting information S2 Table).

A comparison of worker types within the Asian weaver ant colony—minor workers (MW), intermediate workers (IW2 and IW3), major intermediate workers (MIW), and major big workers (MBW)—was conducted based on various body dimensions, including head width, head length, thorax length, body length, and abdomen length (Table 1).

Due to the data's non-normal distribution, caused by extreme values (see Fig 8), the non-parametric Kruskal-Wallis test was used instead of the parametric Analysis of Variance (ANOVA). The Kruskal-Wallis test, which does not require the assumption of normality, assesses whether the medians of the worker types differ significantly based on body dimensions. A p-value less than the significance level indicates a significant difference. The results of the Kruskal-Wallis test were further supported by mean plots (Fig 9), providing a graphical representation of the comparisons. A Box-Cox data transformation did not improve the data normality distribution, therefore, data transformation is not necessary (Supporting information S3 Table). Furthermore, based on the descriptive statistics, it was found that the skewness is between −0.5 and 0.5 (Supporting information S4 Table). The values for asymmetry and kurtosis between −2 and + 2 are considered acceptable to prove normal univariate distribution [28]. Hair et al. [29] and Bryne [30] argue that data is considered to be normal if skewness is between – 2 and + 2, and kurtosis between −7 and + 7 (Supporting information S4 Table). Hence, the data can be considered normally distributed. When the normality assumption is relaxed, the relationships between the dimensions of ant sub-castes are examined using Pearson correlation analysis. This method quantifies the strength and direction of the relationship between variables, producing values between −1 and 1. A value closer to 1 indicates a strong positive correlation, while a closer to −1 indicates a strong negative correlation. Values near zero suggest weak correlations. This analysis highlights the interdependence between the variables. A comparative analysis between the Spearman and Pearson correlation coefficient shows only minimal differences (Supporting information S5 Table).

Based on the correlation analysis, both simple and multiple linear regression models were fitted to the quantitative data. The ant sub-caste groups were excluded from these models, as they require only quantitative variables. The response variable, body length (BL), was chosen, while the predictors included head width (HW), head length (HL), thorax length (TL), and abdomen length (AL). Simple linear regression was performed for individual predictors, while multiple linear regression was applied using combinations of two, three, and four predictors. The best-fitting model was determined to be the full model:

BL = 0.6186 + 1.8020(HW) - 0.0356(HL) + 0.6610(TL) + 1.2459(AL).

Multicollinearity was assessed using the Variance Inflation Factor (VIF) to detect potential redundancy among predictors (Supporting information S6 Table) [31,32]. The VIF values ranged between 5 and 10, indicating moderate multicollinearity in most models. However, this level of multicollinearity is considered reasonably acceptable up to a value reaching 40 [33].

Principal Component Analysis (PCA) was performed to identify which morphometric variables—head width, head length, thorax length, abdomen length, and body length contributed most to differentiating the worker types. PCA reduces the dimensionality of the original variables, summarizing the data through eigenvalues and eigenvectors, and visualizing it using scree plots, biplots, and PCA plots. This technique extracts key information from multivariate data and expresses it in a smaller set of principal components with minimal loss of information. PCA is beneficial when variables are highly correlated, as it reduces redundancy and condenses the original variables into fewer, more informative components. In this study, the Generalized Estimating Equations (GEE) model is also employed, as the ant sub-caste data is clustered, some do not satisfy normality assumptions, and exhibit significant correlations. GEE is an extension of the Generalized Linear Model (GLM) and is more efficient than generalized linear models (GLMs) in the presence of high autocorrelation. It accounts for correlations within clustered data [34,35]. The parameters of the GEE model are estimated using quasi-likelihood estimation, focusing on population-averaged (marginal) effects rather than individual-level effects. Unlike GLM, GEE incorporates a correlation structure, making it more suitable for this type of data. For the GEE model, the response variable is the ant sub-caste (MBW, MIW, IW3, IW2, MW), and the predictors are their body dimensions (HW, HL, TL, AL, BL).

A Generalized Linear Model (GLM) was used to examine the removal of smaller-sized workers in association with the higher daily air temperature (AT°C) at the onset of the foraging activity peak periods [11]. GLMs are flexible regression models that allow for the analysis of response variables following various distributions. Simple and multiple linear regressions are considered the simplest forms of GLM. In this case, the model was considered significant if the p-value was less than the significance level. We assume that both AT (°C) in function with the time interact with the

major workers decision-making in removing smaller-size workers during the identified maximum critical thermal limit $CT_{max}$ (by estimating that major workers are aware of the risk faced under much higher AT °C). The intermediate worker 3 (IW3) and intermediate worker 2 (IW2) removal from the trunk and ground trails was defined as the dependent variable. The onset peak daily foraging activity periods was the independent variable.

A GLM was used to examine the relationship between colony age (younger vs. older colonies) and the number of reproductive queens. A similar GLM analysis detected associations between queens monthly emergence and environmental factors, such as rainfall interception (RI, mm/h) and relative humidity (RH, %).

The Welch two-sample t-test was used to compare the average queen production between younger and older colonies. This test determines whether there is a statistically significant difference between the means of two groups by comparing their means and accounting for variability in the data. The t-test results are considered significant if the p-value is below the specified significance level.

All statistical analyses were conducted using the software R package version 4.3.2 [36].

## 4. Results

### 4.1 *Oecophylla smaragdina* colony social structure description

Figs 4A–4H, 5A–5E, and 6A–6E show how diverse individuals make up a colony. Within each colony structure, members of the *Oecophylla smaragdina* caste represented unique individuals. Fig 5A–5E show the adult stage of each caste, including adult workers and the gravid dealate queen, drone males, winged green, and newly emerged yellow queen. The captured dealate founding queen (Lahad Datu, Sabah MPOB plantations) had a larger abdomen (green alate virgin queens being much smaller). When the alate female first emerged, it had a yellow color and a slightly smaller abdomen. Within a week, it changed to a green hue. In the MPOB plantations in Saratok, Sarawak, mature alate males are smaller than the queens (black-like house flies). The males live in several nests in large numbers (more than 200 for two nests). All other locations' nest collections revealed a lack of adult male presence. The workers' sub-caste comprised individuals with barely discernible color contrasts and noticeable differences in their abdominal and body sizes (five different sizes). The individuals were the major big workers (MBW), larger and darker reddish-brown; the major intermediate workers (MIW), intermediate workers 3 (IW3) and 2 (IW2); and the minor workers (MW) (Fig 5A–5E). Among workers, the three novel sub-castes were identified (Fig 5E, red and yellow circles). The considerable variation in body length between each worker is depicted in Fig 5E. The large major workers' abundance is noticeably greater than for minor workers in all collected nests from various colonies and geographical sampling locations, but more balanced with the two intermediate workers 3 and 2.

**4.1.1 Colonies' egg count estimation.** Table 2 shows the ANOVA summary for the average number of eggs inside 8.34 mm x 5.02 mm standardized-size clusters from Felda Gunung Besout Perak oil palm plantations (Supporting information S1 Table). The total egg production per colony is estimated to fluctuate from an average of 16,440.00 (younger colony) to 41,040.00 (older colony). The corresponding period taken by an egg-laying queen to produce such an amount is unknown. The colonies did reveal significant differences in egg production as shown by the Post-hoc Tukey HSD test for multiple comparison of the means (Supporting information S2 Table).

**4.1.2 Colony caste composition-description.** Eggs recently laid by queens were white to slightly yellowish and formed into cubic clusters surrounded by an unidentified yellow gelatinous substance. The egg's sharp thin tip corresponds to the future head of an individual ant (Fig 4A, black circles). All captured brood nests contained these egg clusters in distinct whitish silken compartments on the nest's underside. At a later time, the embryos were white and had a distinct miniature oval shape. Eggs are smaller than the characteristic white, elongated worm shape of the larvae, without body parts growth. The width varies between 0.37 mm and 0.50 mm, while the length varies between 0.96 mm and 1.15 mm (N = 50). Eggs are always found as compact, clustered, shiny, adhesive entities (Figs 3A, 3B, 4H), whereas larvae and pupae are found separately as dry, solitary, non-sticky entities. (Fig 4B and 4H, red circle). Fig 5A–5H

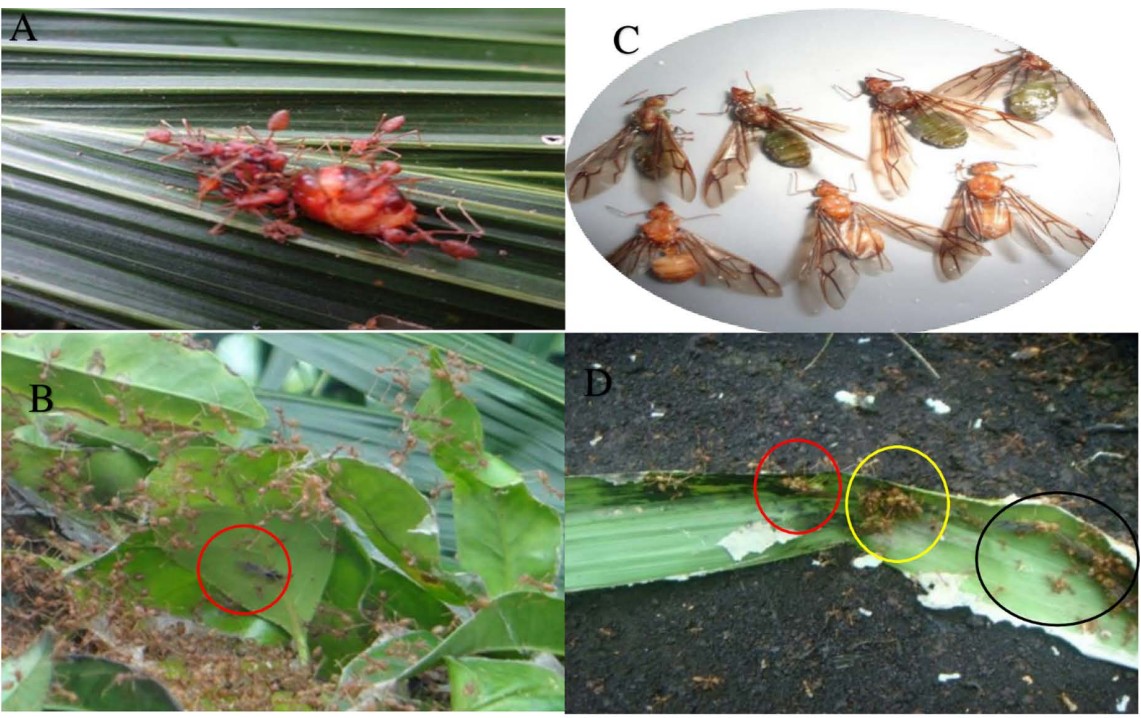

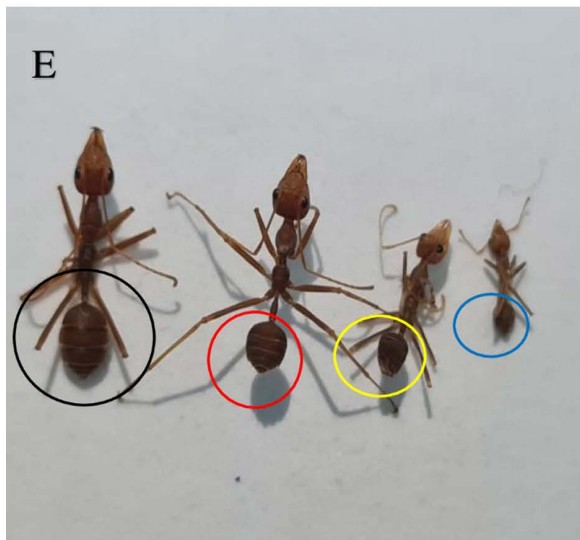

**Fig 5. Various castes of *O. smaragdina*. (A)** Gravid dealate founding queen with large gaster surrounded by major workers; **(B)** Drone male (red circle) among major workers; **(C)** Winged green virgin queens & yellow callow; **(D)** Dealate gravid queen (red circle) & another wingless queen surrounded by major workers (yellow circle); Multiple winged queens (black circles), major workers-minor workers & larvae-pupae exhibiting a sharp white color; **(E)** Large major worker IV (Black circle); intermediate worker III (red circle); intermediate worker II (yellow circle) and minor worker I (blue circle). Photo Credit (Moïse Pierre Exélis).

depict the immature stages of the various castes that make up the social structure of the *O. smaragdina* colony, which consists of eggs, workers, the queen, male larvae, and queen and worker pupae. The eggs are clustered, encased in an unidentified protective substance, and much smaller than larvae (Fig 4A, separated under 80% ethyl alcohol). Larvae are

whitish, more elongated than eggs, and have a thinner, sharper extremity that corresponds to the head and resembles the shape of worms (Fig 4B, 4E, 4H). Pupae (Fig 4F) are distinguished from larvae (Fig 4E) by a foetal position displaying the appearance of eyes, mouth, legs, and wing growth that accurately predicts the shape of the adult weaver ant. Queen and male larvae are distinguished by their interior curves' lighter and darker hues, respectively (Fig 4D).

**4.1.3 Colonies dissected nest content – population size estimation.** Due to the impossibility of distinguishing visually between queen, brood and barrack nests, only two categories are considered. The height of the brood and the barrack nest made it simple to differentiate between the two. Depending on the extent of the palm tree, brood nests were located at heights between 6 m and 10 m, whereas barrack nests were typically located between 2 m and 4 m. The field's dissected barrack nests (N = 50) were inhabited by larger-sized dark brown to reddish MBW and MIW sub-caste (Fig 6A, 6B). In Felda I colony nest samples, queens, male larvae, and pupae were collected (male adults absent; Fig 5C, 5D, 5E, 5G). As described in Section 4.3, all caste forms are represented (see Fig 5A–5H). An estimate of the reproductive winged queen population size ranged from 9 to 11 per nest in early matured colonies (24–36 months of age; N = 10 colonies), yielding an average range of 63–156 queens per colony. In older matured colonies (48 months and above), alate queens range from 19 to 21 per nest and 475–630 queens per colony. In brood nests with extensive egg clusters, yellow larvae of an unidentified species of large homopterans were discovered (collected first in May 2012 at MPOB plantations in Teluk Intan, Perak). The yellow larvae exuded a transparent nectar-like substance from their midline. These homopterans were cared for by major workers who were observed gathering these ingredients (under laboratory conditions). The contents of *O. smaragdina's* nests in the Felda I colony are analyzed descriptively (Supporting information Tables 2 and 3). A single nest sample result for UM III large-size colony had a total worker population of 64,761 individuals. The presence of reproductive adult males was not detected in the samples. We managed to sample male pupae and larvae in the younger colony. Many adult males were collected from guava trees *Psidium guajava* L. (Myrtales: Myrtaceae) near occupied palms, secluded in two separate nests in Borneo Sarawak (young colony). Significantly more major workers exist than intermediate or minor workers. This colony's average number of workers was estimated to be 1.02. Million. Another large population of Asian weaver ants was discovered in MPOB plantations in Saratok, Sarawak, with 149 interconnected polydomous nests occupying 12 palms.

An estimation of the population size of Felda I is: $PS_{Felda\ I}$ = 33 nests x 11,962.00 = 394,746.00 ± 159,647 and UM III, $PS_{UMIII}$ = 74 nests x 13,761.00 = 1,018,314.00 = $1.02 \times 10^6$ ± 525,136 representing an average of major-intermediate-minor total workers.

**4.1.4 Workers sub-caste description.** Fig 6A–6E shows the size variations between the main body parts, including HW, HL, TL, AL, and BL, with the abdomen and body length showing the most notable variation. There are 12 antennae segments for all the workers (Fig 7, black circle). The data distribution shows consistency for all five identified sub-castes except for the major big worker (MBW) thorax and intermediate worker 2 (IW2) head width (Fig 8). The MBW head width and length are consistent variables that define a significant difference from other workers. MW-HW, HL, and BL are consistent (less variability) except for TL and AL. IW2-HW is inconsistent (high variability). IW3 TL is more variable than IW2. MBW -TL, BL, and Al are inconsistent (high variability).

## 4.2 Workers' morphometric measurement analysis

**4.2.1 Kruskal-Wallis rank sum test.** There are differences in length and width dimensions between all the worker types. The samples do not have overlapping 95% confidence intervals, the Kruskal-Wallis rank sum test indicates a significant difference among the five sub-castes of workers (MBW, MIW, IW3, IW2 & MW) morphometric variables BL, AL, TL, HW, and HL (Fig 9).

**4.2.2 Correlation and clustering.** All variables have a strong positive correlation of more than 0.9 for HW-HL, HW-TL, HW-AL, HW-BL, HL-TL, HL-AL, HL-BL, TL-AL, TL-BL, and BL-AL and they are all significants (Fig 10).

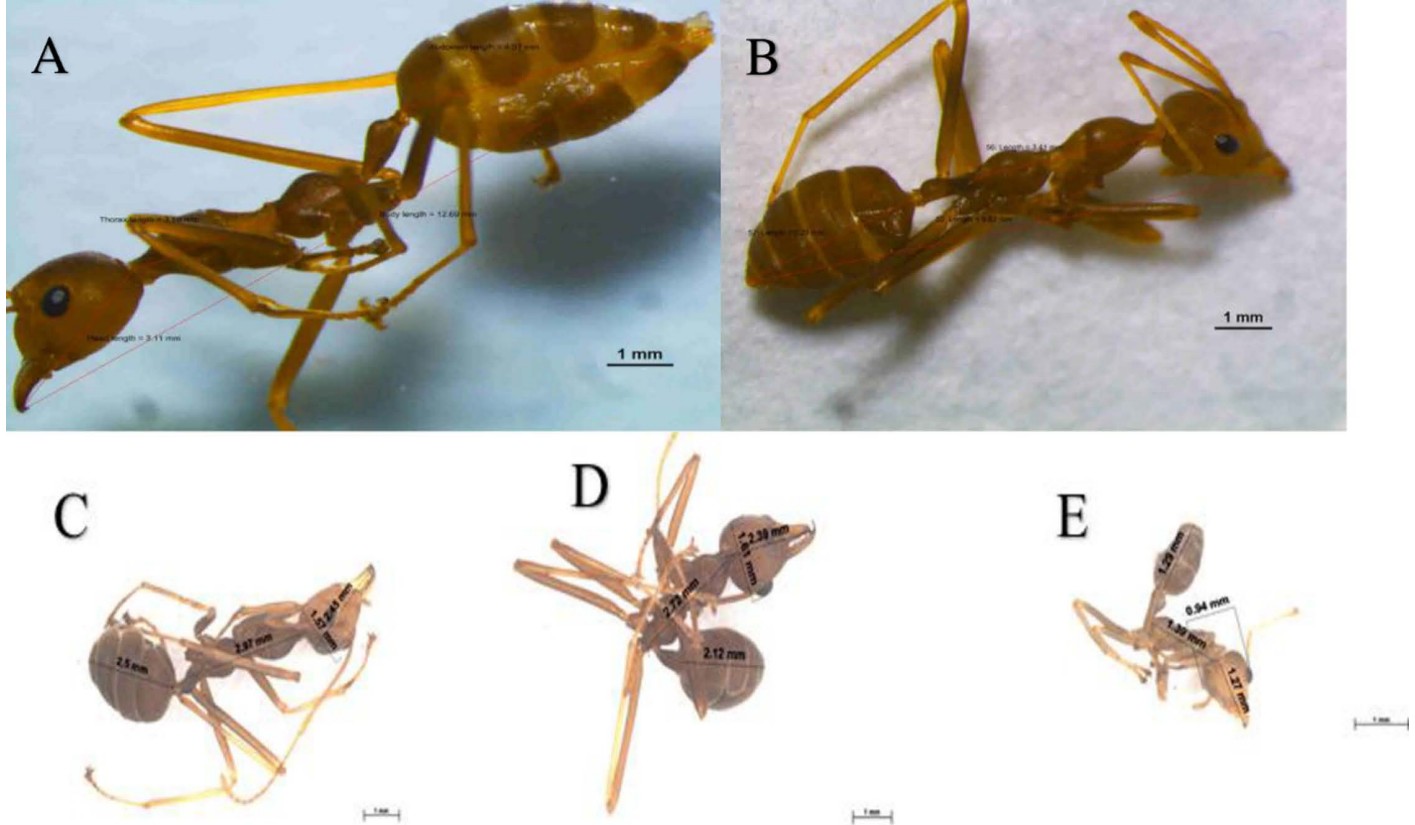

**Fig 6. Distinct worker sub-castes. (A)** Major big worker. **(B)** Major intermediate worker. **(C)** Intermediate worker 3. **(D)** Intermediate worker 2. **(E)** Minor worker. Examination by stereomicroscope Nikon SMZ800N (A-B) and ZEISS Stereo Discovery V20 microscope **(C-E)**. The lines indicate the sizes of variables HW, HL, TL, AL, and BL. Photo Credit (Exélis Moïse Pierre).

**Table 2. One-way ANOVA post-hoc Tukey HSD.**

| Source of variation | SS | df | MS | F | P-value |
|---|---|---|---|---|---|
| Colony | 24464.20 | 9 | 2718.24 | 21.62 | $2.06 \times 10^{-5}$ *** |
| Residuals | 1257.00 | 10 | 125.70 | | |
| Total | 25721.20 | 19 | | | |

Signif. codes: 0 '***' 0.001 '**' 0.01 '*' 0.05 '.' 0.1 ' ' 1.

Based on the eigenvalue = 4.77 (greater than 1), Dim 1 (PC1) shows a significant variation. PC1 can explains that a high proportion of variance for PC1 contributes 95.5% of the total variation in the dataset (Table 3). The scree plot (Fig 11A) supports the numerical analysis based on the eigenvalues, Dim 1 (PC1 = 95.5%).

From the PCA variable plots (Fig 11B – left & right) and biplot (Fig 11C), the angles between TL, HL, and HW variable vectors are acute, indicating that the principal components contain less information about these variables. AL and BL variable vectors are the longest (red and green arrows, respectively), indicating that the principal components contain significant information about these variables. Angles between TL, HL, and HW variable vectors are small (almost zero), indicating that the variables are collinear (lie on the same line, the direction is similar). Angles between AL and BL variable

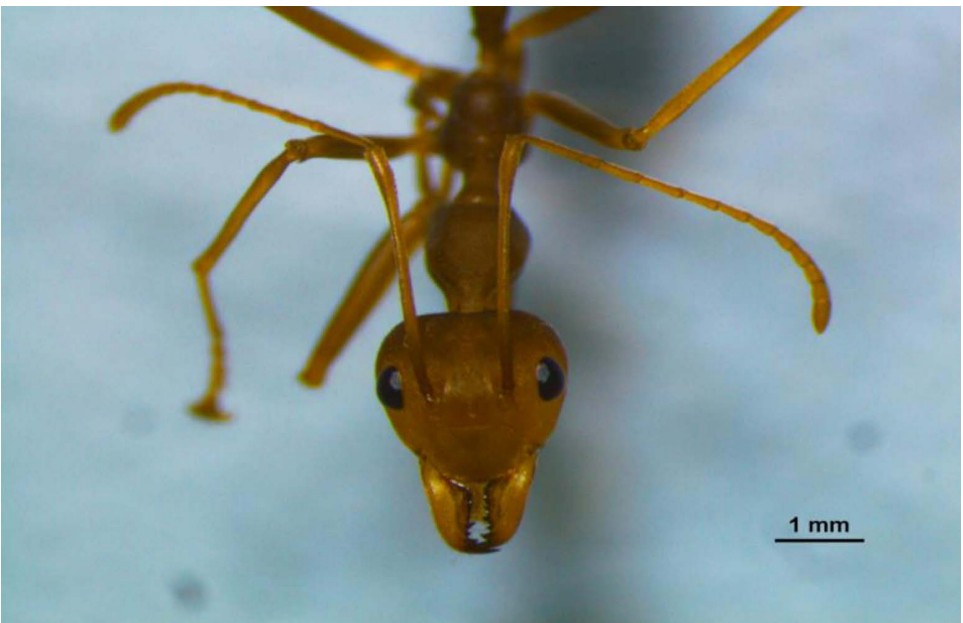

**Fig 7. *O. smaragdina* worker Antennal segments.** Examination by stereomicroscope Nikon SMZ800N. Photo Credit (Exélis Moïse Pierre).

vectors are less than 90, showing a strong correlation between the variables. Fig 11C–11D demonstrate that the ratings of the abdomen and body length (AL, BL) have greater significant variability than the measurements of the head width, head length, and thorax length (HW, HL, TL), which include all five different worker group clusters. Fig 11C–11D show how each variable differs among the five significant clusters (MBW, MIW, IW3, IW2, and MW). There are statistically significant differences between all five worker classes, primarily dependent on the length of the abdomen and body.

**4.2.3 Generalized Estimating Equation (GEE).** The results show that the p-value $< 0.05$, the variables are statistically significant (Table 4). There is strong evidence that variables significantly affect the response. From the results, Head length (HL), Thorax length (TL), and Body length (BL) influenced the ant worker sub-caste classification.

The model is

$$\text{Type} = -7.016 - 1.632\{HW\} - 4.281(HL) - 1.494(TL) + 0.606(AL) + 1.234(BL)$$

### 4.3 Major – intermediate workers' behaviors

The two sub-castes of intermediate workers do not function as nurses, as do minor workers, but rather exhibit the same behaviors as large major workers (i.e., defensive guard duty, aggressive, exploration of territories). They have been observed patrolling extensively with major workers along the trails at the primary trunks, fronds, and rachis of palm trees (Fig 12). Fig 12 depicts an intermediate worker 2 (red circle) foraging alongside an intermediate worker 3 (yellow circle) on a palm tree frond beside a major intermediate worker (black circle) in Felda Gunung Besout plantations. These individuals were observed foraging on the ground and monitoring the trunk bases of palm trees. They also react aggressively by opening their mandibles widely and raising their gaster to a defiant 90-degree angle before assisting in the attack of intruders, along with larger major workers. They are positioned behind the first line of repulsion (controlled by the major big worker and major intermediate worker) against alien elements in their territorial interaction zone. Their occurrence outside the colony's territorial perimeter is less frequent than that of larger-sized major workers. There is a suggested association with varying daily weather parameters, such as air temperature and atmospheric pressure, in the absence of

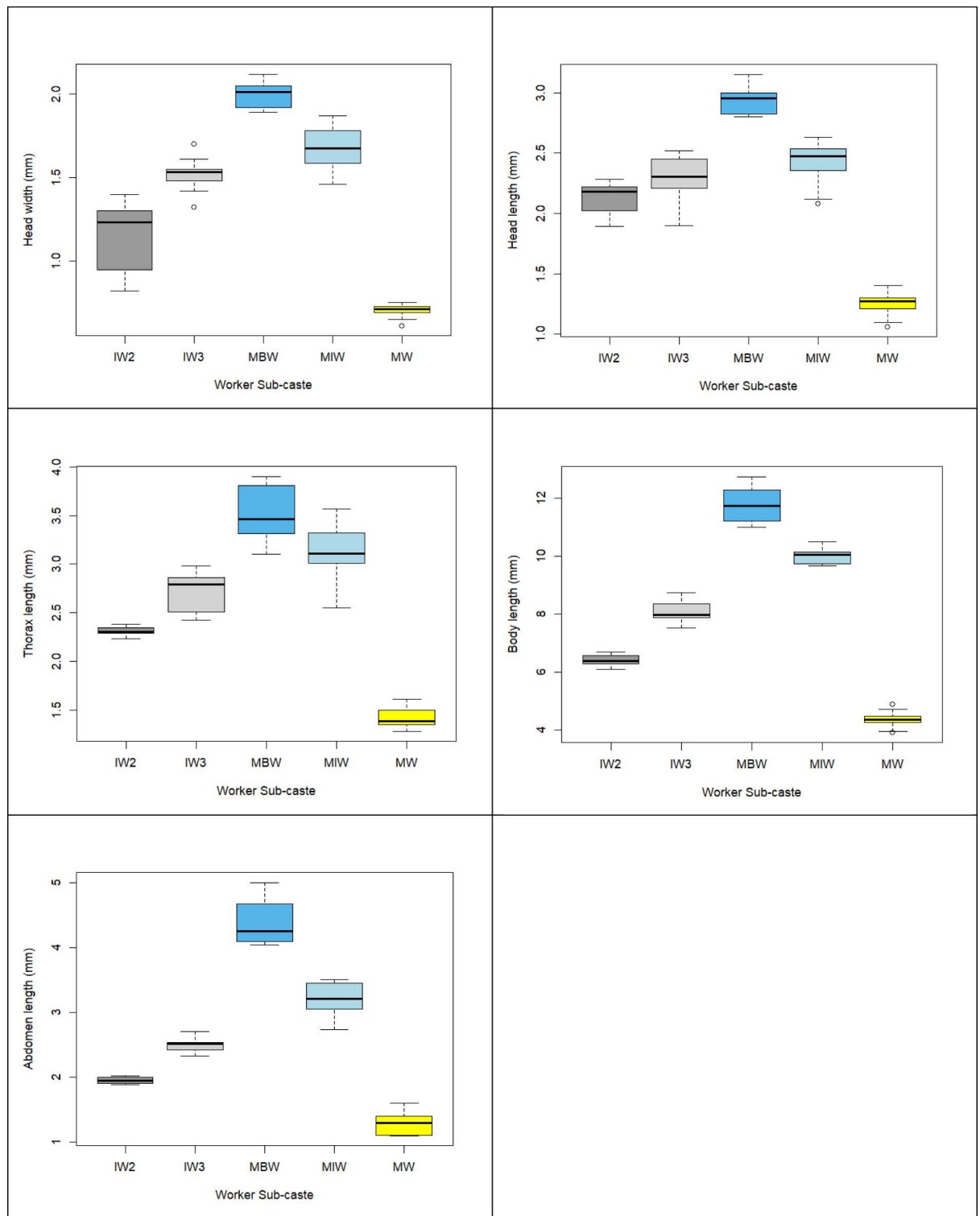

**Fig 8. Box plot: Comparison of body dimensions (HW, HL, TL, BL, AL) by worker sub-caste.**

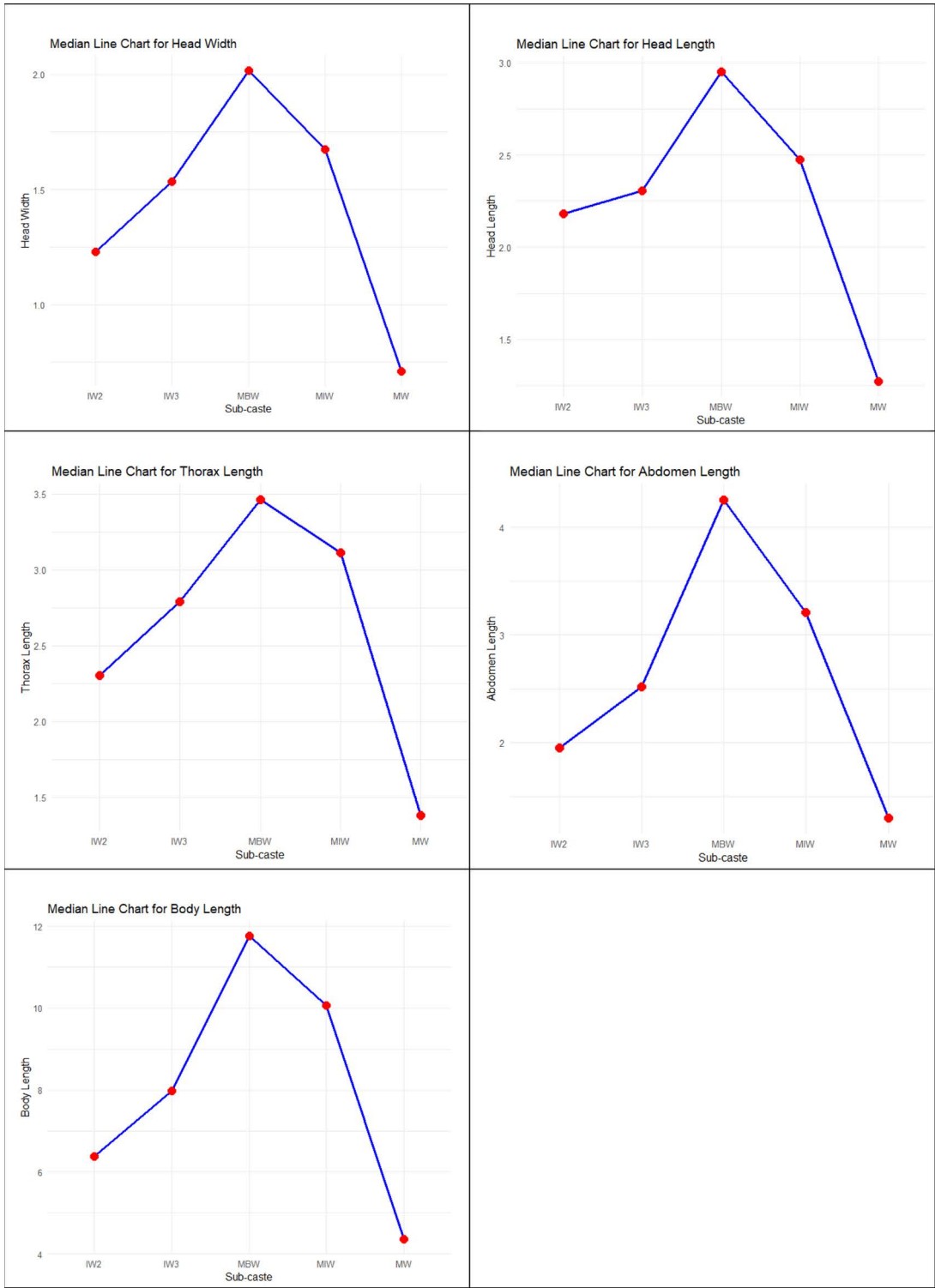

**Fig 9. Kruskal-Wallis median plot.** The median line plots are generated from the Kruskal-Wallis test. These plots illustrate that each sub-caste of ant workers has significantly different medians for body dimensions (HW, HL, TL, AL, BL), showing how the median of each morphological trait varies across different ant sub-castes.

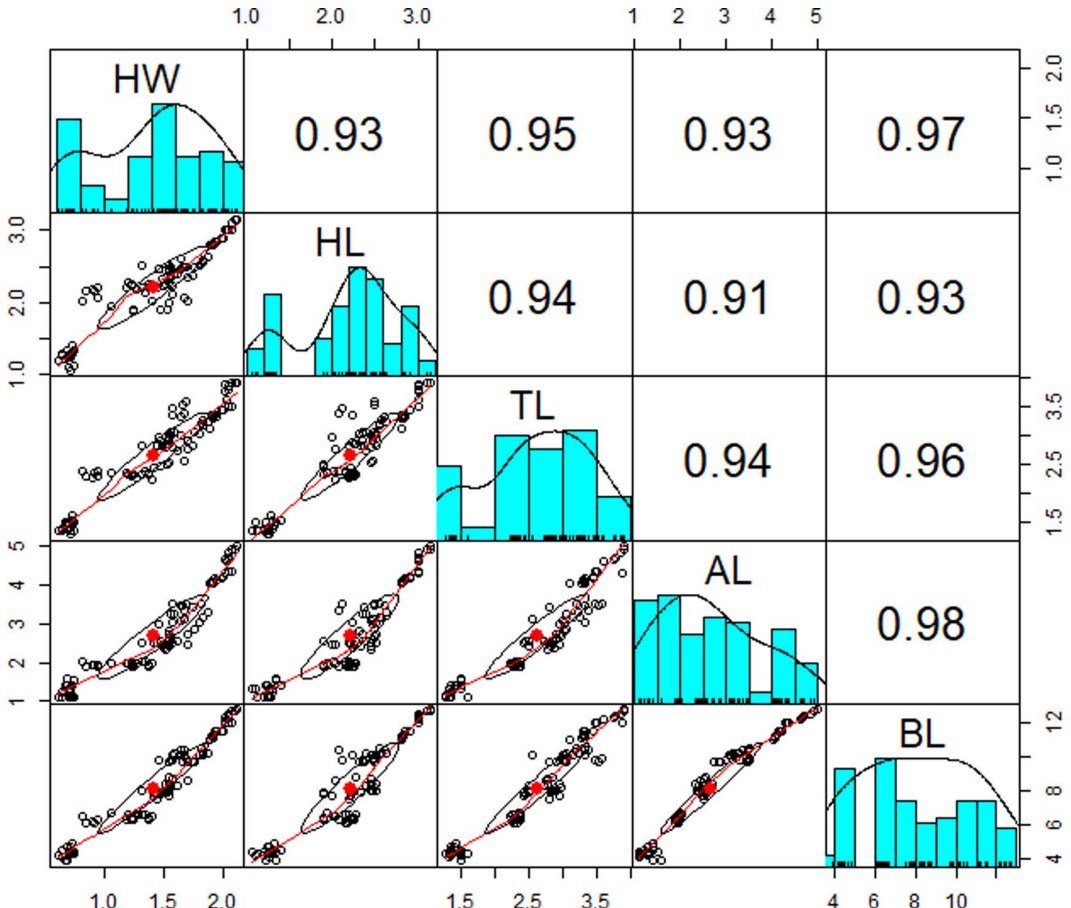

**Fig 10. Correlation plot (Pearson correlation – histogram).** The correlation plot illustrates the relationships between body dimensions. The corresponding correlation values quantify the strength of each relationship. The closer a value is to one, the stronger the positive relationship between the variables, namely HW, HL, TL, AL, and BL. The red spots represent the mean values.

**Table 3. PCA Eigenvector proportion – variables contribution.**

| Variables | PC1 | PC2 | Eigenvalue[a] | Standard deviation | Variance proportion[b] (%) | Cumulative variance (%) |
|---|---|---|---|---|---|---|
| HW | −0.4475 | −0.0477 | 4.7746 | 2.1851 | 95.4900 | 95.4900 |
| HL | −0.4408 | −0.7499 | 0.1045 | 0.3233 | 2.0900 | 97.5800 |
| TL | −0.4489 | −0.1032 | 0.0600 | 0.2450 | 1.2000 | 98.7800 |
| AL | −0.4456 | 0.5662 | 0.0470 | 0.2170 | 0.9420 | 99.7260 |
| BL | −0.4529 | −0.3223 | 0.0137 | 0.1170 | 0.2740 | 100.0000 |

[a]Eigenvalues > 1 are preferred.

[b]PC1 can explain 95.5% of the total variation in the data set.

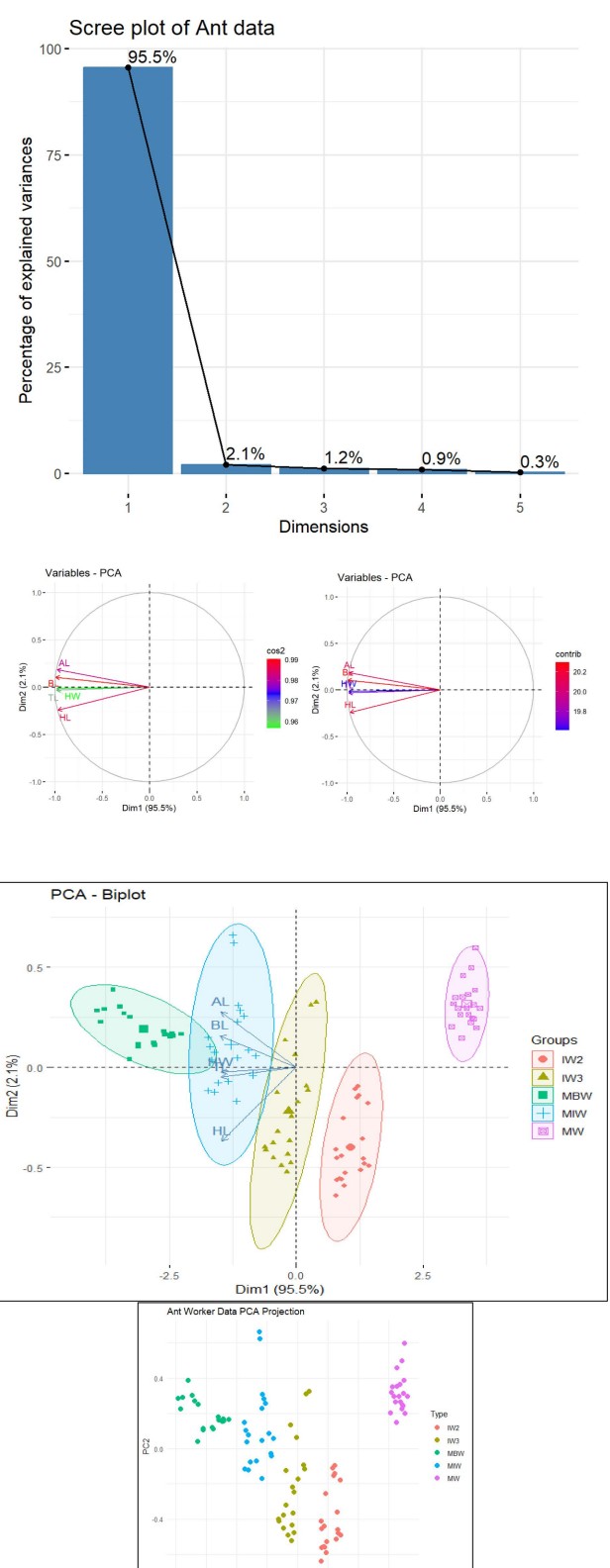

**Fig 11. A) From the scree plot, PC1 explains the most variance (95.5%), meaning it captures most of the variation in the dataset.** PC2 explains around 2.1%, adding important but less dominant variation. PC3, PC4, and PC5 contribute much less (0.3%−1.2%), suggesting they can be ignored.

B) PCA variable plots. Based on the arrows, all variables contribute almost equally to the principal components. HL, BL, and AL (in red) have high cos2 (cosine squared) values, meaning they are well-represented. HW and TL contribute less to the selected principal components. HW and TL are closely aligned, so they are highly positively correlated. The arrows are pointed to the left, which indicates that all variables (HW, HL, TL, AL, BL) have negative loadings on Principal Component 1 (PC1). Higher PC1 scores correspond to lower values of these original features (B – left). HL, BL, and AL (in red) have high contribution values influencing the components. HW and TL contribute less to the selected principal components (B – right). HW and TL are closely aligned, so they are highly positively correlated. HL, BL, and AL are positively correlated (B – left and right). **C-D)** Analysis plots: Biplot – PCA projection plot. Biplot: MW tends to cluster separately. MBW and MIW overlap slightly but still form distinguishable groups. Hence, PC1 and PC2 can differentiate all the ant sub-castes well. The plots (Biplot & PCA projection plot) show that the ant sub-castes are clustered and well-defined.

**Table 4. Generalized estimating equation.**

| Predictor | HW | HL | TL | AL | BL |
|---|---|---|---|---|---|
| P-values | 0.0910 | 0.0000 | 0.0010 | 0.0672 | 0.0000 |

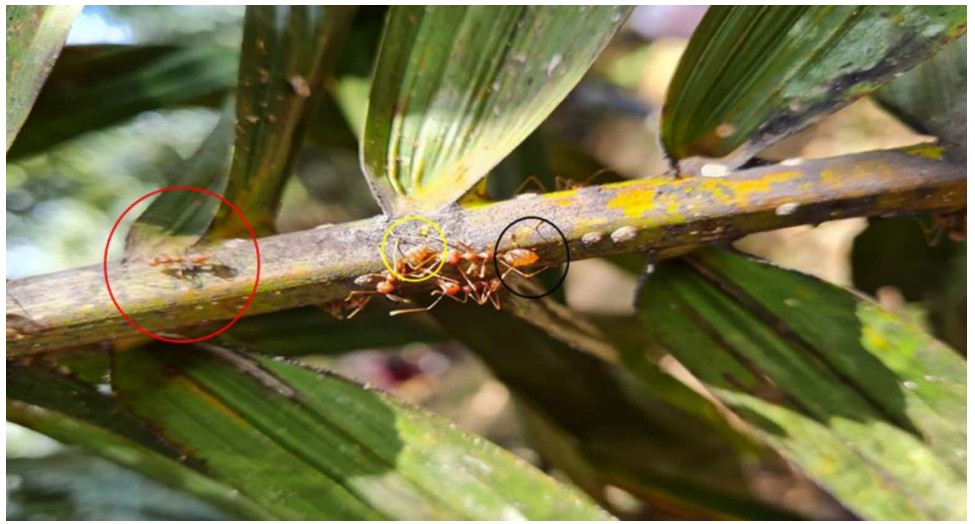

**Fig 12. Major and intermediate workers' collective daily foraging activity.** Various workers are in daily activity on a palm frond: Major intermediate worker (black circle); intermediate 3 (yellow circle); intermediate 2 (red circle). Photo Credit (Exélis Moïse Pierre).

precipitation. However, the intermediates have been spotted acting on solitary foraging or by a small group of two to three individuals, exploring far from their territorial boundary during lower temperatures.

Similarly, during the daily hot hours (average: 1200–1700 hours), major intermediate workers and major big workers were frequently observed acting as supervisors to limit the activity of intermediate workers 3 and 2 far from nest canopies by bringing them back to nearby nests by holding them at the thorax baseline with their mandibles. During the hottest daily period (1100–1600 hours), the number of foragers of intermediate size active was severely restricted, as was their ability to travel beyond nest perimeters. The major workers acting as leaders or supervisors impose an embargo on their foraging activity within the oil palm trunks and the ground. The intermediate workers are systematically caught and sent back to their nests (Fig 13A). The GLM results show that the simple and multiple regression have a linear relationship between the number of removed intermediate workers and the higher daily air temperature, exposed an association with periods (p<0.01), and a strong positive Pearson correlation (0.82 – Fig 13B). Generally, intermediate workers refused to comply with the imposed movement temporary embargo (relentlessly joining the trunk and ground trails). Intermediate workers are still free of movement within the canopies under their heat-protective shadows.

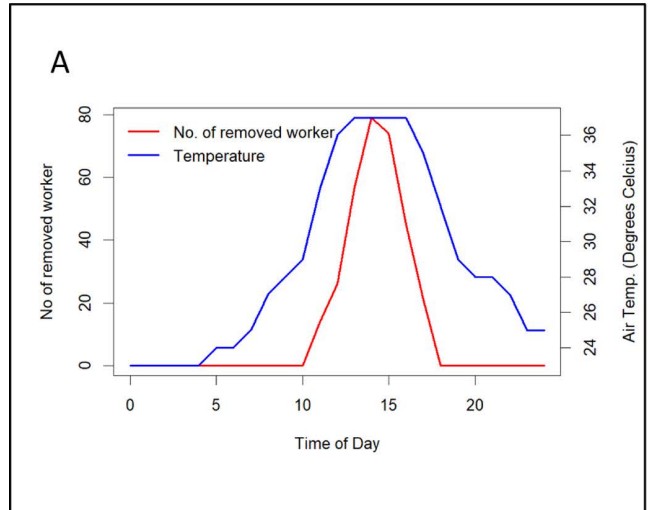
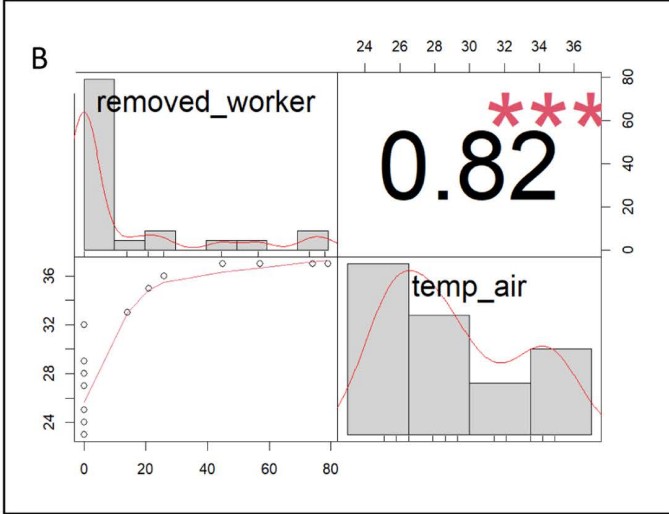

**Fig 13. Leadership and supervision activity of bigger size major workers.** Removal of intermediate workers 2 and 3 during the higher AT (°C) periods (1150 hour to 1600 hours – A - left) by the major workers. Pearson correlation significant coefficient (B - right).

### 4.4 Emergence-abundance of reproductive individuals

Every month of the year, samples of reproductive individuals were taken, but their dynamic was not assessed. It was made up of adult yellow- and green-winged queens. In the MPOB plantation in Saratok, we sampled drone males once and larvae-pupae in Felda Perak (Supporting information S7 and S8 Tables). In oil palm plantations, it is rare to see adult males. The bar plot (Fig 14) demonstrates that *O. smaragdina* reproductive emergence occurs throughout the year, with an increase in rainfall interception (RI mm/h) and relative humidity (RH%) levels. May through August saw fewer mature or immature queens, correlated with lower monthly rainfall interception and drier days. October through February saw greater monthly rainfall interception, which led to a rise in reproductive emergence. A brood nest located at 10 meters high had a massive concentration of mature green-winged queens that was exceptionally much higher than in other months. The GLM results show that the simple and multiple regression have a linear relationship between the number of emerging queens and rainfall-relative humidity ($p < 0.05$). A strong positive correlation under Pearson exists between the total number of queens and RI mm/h – RH% (Fig 14).

The average number of reproducing queens and males in both young and older colonies, per nest and colony, is depicted in Figs 15A–15B and 16A–16B. The queen abundance is significantly higher in older colonies (mean number), while the production of adult males (mean number) decreased in the older matured colonies. The mean number of queens in the older colony is more than in the younger colony ($t = 6.8433$, df $= 13.88$, p-value $= 4.2 \times 10^{-6}$). The mean number of males in the older colony is less than in the young colony ($t = -8.7963$, df $= 9.9507$, p-value $= 1$) (Fig 15A–B – left represents younger and right older colonies).

## 5. Discussion

### 5.1 *O. smaragdina* colony social structure – Novel intermediate worker's caste

The establishment of a colony of Asian weaver ants is contingent on the successful mating of drone males and winged virgin females (potential future queens) during a nuptial flight swarm [23,37]. Once inseminated, mated alate queens seek a suitably curved leaf to lay the first egg batch in a "claustral" state and shed their wings. This corresponds to a state of self-sufficiency by confinement, with the queen relying on her reserves to care for the brood until their

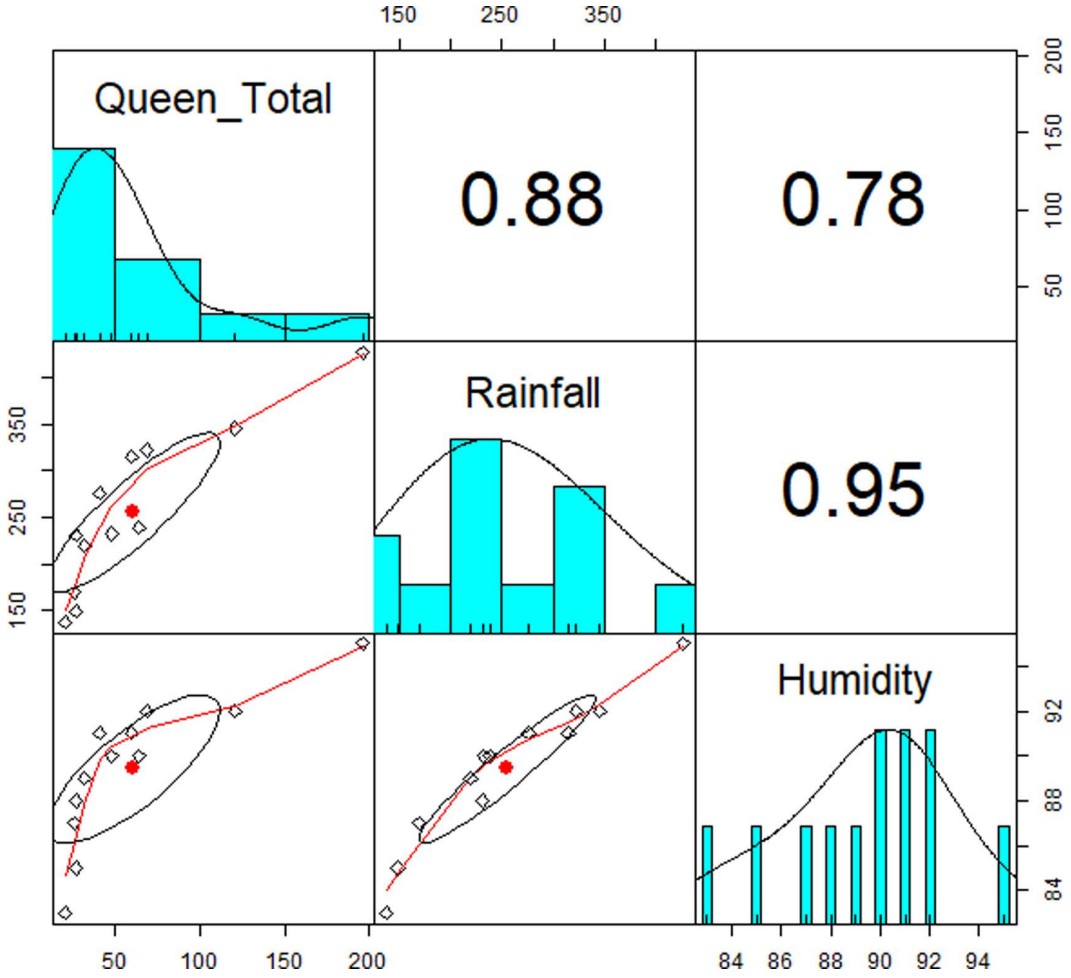

**Fig 14. Correlation plot (Pearson correlation – histogram).** The correlation plot shows the relationship between the number of emerging queens, rainfall intensity (mm/h), and relative humidity (RH%). There is a strong positive correlation between the total number of queens, rainfall intensity, and humidity. As rainfall intensity and humidity increase, so does the number of queens. The red points represent the mean values.

emergence) [5,23,38]. The first generation consists solely of major workers tasked with foraging and taking over all activities at the colony level, including care for the brood [21]. When the second egg group produces a mixture of major and minor workers, colony expansion commences with the completion of the nest. At this point, main workers use late instar larvae as a shuttle to seal the nest with silk [1,5]. In most reported studies, workers of the Asian weaver ant exhibit typical dimorphism or a bimodal size-incidence distribution characterized by the major and minor workers [1,9,21]. This study suggests the existence of a polymorphic reality by reporting for the first time a "polymodal size" distribution among five distinct worker inter-class or sub-castes comprised of two major, two intermediate, and minor individuals. Masram & Basargade did mention structural workers' size differentiation and a more complex condition, but failed to provide unambiguous data on the existence of additional sub-castes of Asian weaver ants [39]. The study mentioned the existence of *O. smaragdina* polymorphs without distinguishing in-depth distinct worker sub-castes (no data on main body parts size variations) but based solely on leg structure and sensilla size and instead provided a description of sensilla among all worker castes [39,40]. Another study analyzing the biochemical secretions (i.e., formic acid, free amino acid, and acetylcholine) of all castes in India discovered that *O. smaragdina* colonies are polymorphic [15]. Workers of *Oecophylla*

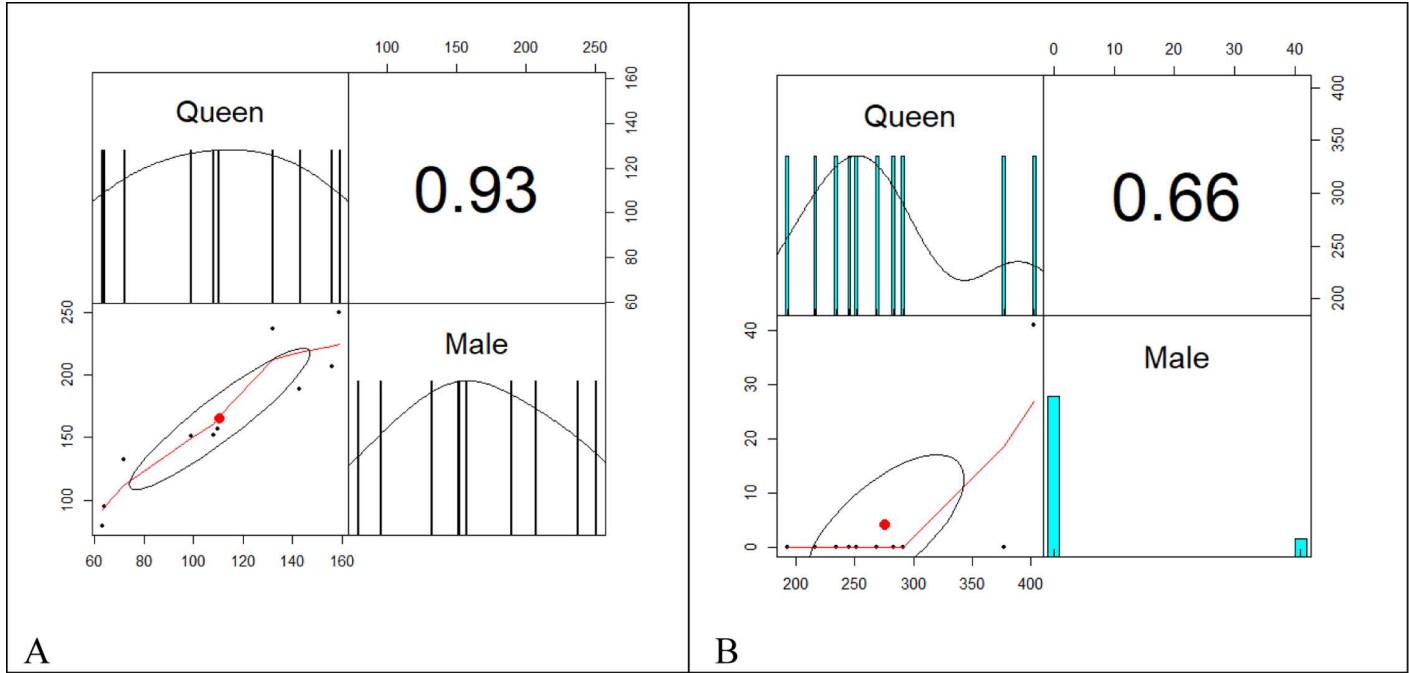

**Fig 15. Correlation plot (Pearson correlation) of young colony (left) and old colony (right).** From Panel A–B, there is a strong positive correlation between the number of queens and males in the young colony compared to the old colony, respectively. In the young colony, as the number of queens increases, so does the number of males.

*smaragdina* are divided into three distinct sub-castes, namely major, intermediate, and minor, without defining their respective sizes [15].

This study presented the fundamental social colony structure composition of the Asian weaver ant in Malaysia, which is similar to previous studies [5,9,13] except for two novel intermediate workers and larger-sized major workers (darker color) that have never been reported. Pimid et al reported *O. smaragdina* basic colony structure throws the chain of mutation from eggs to immature stages (larvae, pupae) to adulthood emergence [13]. It consisted of reproductive phase individuals (alate virgin queens, dealate shed wings founding queens, and alate males) with non-reproductive ones (workers). However, Pimid et al. referred to the larvae as eggs, although both characteristics are distinguishable (the photograph depicted only larvae, not eggs as stated in the manuscript), as shown in Fig 5H and described in the results section [13]. Following Rwegasira et al. and Van Itterbeeck et al. the reproductive individuals are available throughout the year, particularly on rainy days when relative humidity is higher [41,42]. Acquiring adult male alate samples is far more difficult due to nest occupation segregation between males and the rest of the colony's members. Vanderplank described *O. longinoda* males as inhabiting different nests than the remainder of the colony members, and this report is consistent with our findings [43]. However, the study findings cannot define if this occurs just before the nuptial flight or if it is just their common natural behavior. Despite the difficulty of capturing adult male drones, we can capture their larvae. Male weaver ants were discovered in a peat soil oil palm plantation in Saratok, Sarawak, and Felda Perak. Due to the limited sample size of colonies (n = 10) and the existence of split sex ratios in many ant species this may be a coincidence [44,45]. After sustained rainfalls, alate male lifespan is exterminated immediately following the post-massive swarming nuptial flights during the mating period, which is suggested to be another potential factor [23,46]. It's conceivable that the sampled reproductive male nests with an abundance of green-winged queens represented an aggregation preceding the nuptial flights [23,46,38]. Pimid et al performed brood collection on *O. smaragdina* colonies occupying a smaller area

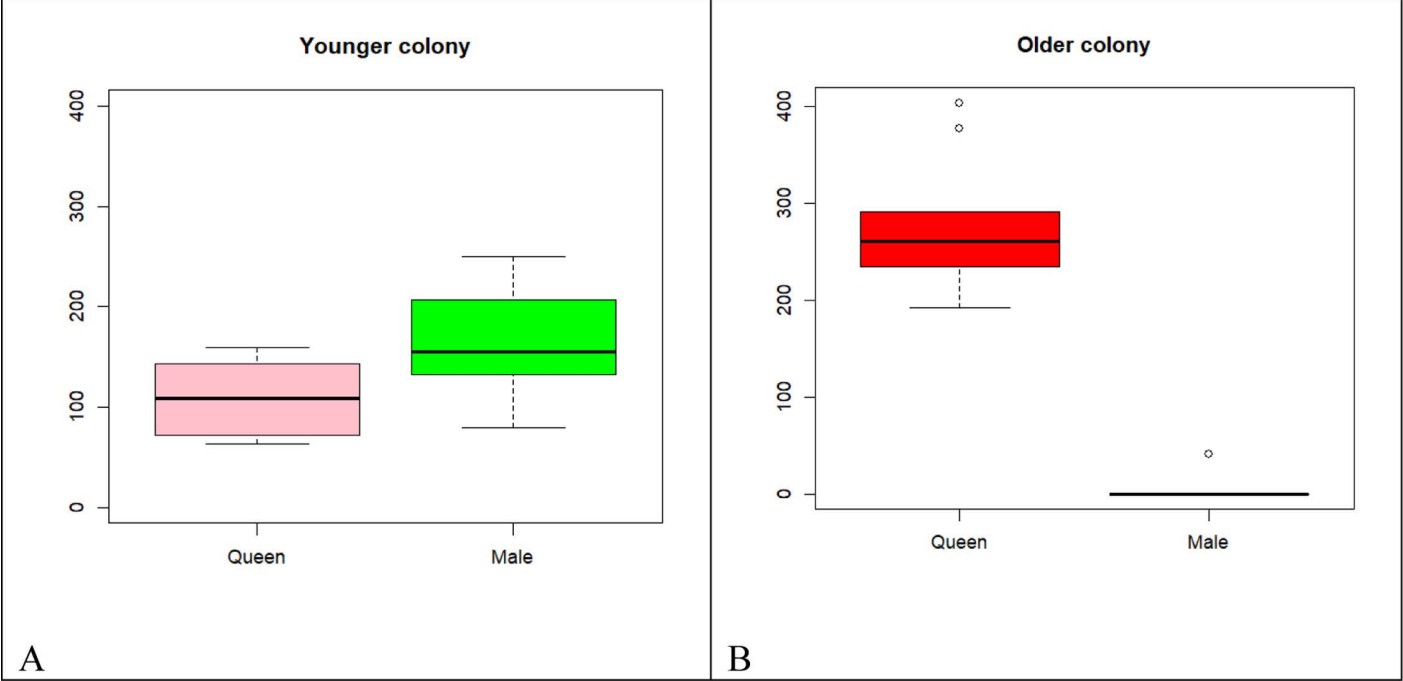

**Fig 16. Box plot of queens and males in the young colony (left) and old colony (right) A-B compares the queens and males distribution in the young and old colonies, respectively.** In the young colony, the median number of males is higher than that of queens. However, in the old colony, the median number of queens is approximately 2.5 times higher than in the young colony, while the number of males has decreased significantly.

(mango and pongame oil trees) on an urban university campus compared to a random sampling conducted in vast oil palm plantation zones [13]. Compared to common palm tree plantations, the probability of sampling adult male nests in restricted occupied spaces from much shorter trees is greater. As suggested for other arboreal ant species by Powell et al. [47] and the Asian weaver ants by Rahim & Okahwara [48], nest spatial distribution and configuration may vary according to topography and tree phenology. In extensive oil palm plantations, therefore, it is difficult to identify the nests of winged males. Our findings concur with those of Holldobler & Wilson [5] and Frank [49], who reported that male production is scarcer in mature, larger colonies than in smaller colonies (Fig 14A–14B). Hasegawa et al [50] insisted that mortality is a prime factor during the colony reproductive stage. In a previous study, Vanderplank conducted dietary experiments on the African species *O. longinoda*, noting the appearance of large, intermediate, and small-sized workers in colonies fed a mixture of protein (insects) and honey diluted with water [43]. However, there is no evidence that the type of diet correlates with workers' caste differentiation, except for variations in aggressiveness and population density [51]. Furthermore this previous study did not provide morphometric data on these workers' distinct intermediate categories, his observations are incorporated into the current study's findings [43]. Consistent with all previous research, there is a greater number of major workers than minor workers [13,52–54]. Foragers have a greater functional spectrum than the rest of the colony. They protect the entire colony perimeter as defensive elements) [5]. They provide many veteran workers with permanent close protection of the founding queen, are responsible for brood transport and nest construction-repair, and search for daily food items [1,27,55]. According to reports, majors also assist drone males during the nuptial flight by escorting them back to the nest in the event of strong winds [38]. Minor workers serving as nurses are primarily confined to the vicinity of the nest, where they care for the nest broods [27]. The existence of novel intermediate workers contradicts the concept of a bimodal size distribution in which major and minor worker sizes do not coincide [5,55]. In incipient colonies, the first yield of gelatinous, sticky eggs and the first generation are susceptible to dehydration from heat [38]. The compartment

chamber arrangement within the brood nests [56] may maintain a cooling effect, thereby preventing the eggs from drying. This is done to avoid the temperature threshold (the highest temperature tolerance level that would inhibit the growth of any ectothermic organism) to maintain larval development [57]. The silken compartments function as protective walls, which may serve to separate reproductive forms from each other's castes. Nuptial flights and mating are preceded by days of rainfall and high humidity before, during, and after [38,46], resulting in much cooler air temperatures and a safer climatic environment [23].

### 5.2 Novel intermediate workers' behaviors

It is evident from field observations and monitoring that intermediate workers also serve a purpose within the colony. Our study indicates that their behaviors are consistent with the concept of labor division in *Oecophylla* weaver ants [56]. However, additional experimental analysis is required to evaluate these novel worker sub-castes. There is no reported evidence of a leadership occurrence among ant individuals [58]. However, the study is the first report showing the occurrence of supervisory acts from major workers acting as leaders controlling the more vulnerable, smaller-sized intermediate workers to remove them from daytime overheat exposure. The systematic removal of smaller-sized workers perceived as vulnerable to the critical thermal maximum ($CT_{max}$) is suggested to be preventive [59,60]. Penick et al suggested that a thermal adaptation trade exists for a smoother brood growth development occurrence [20]. While performing extensive foraging and reconnaissance activity far from nests along palm tree trunks and foliage, intermediate workers displayed territorial behaviors with less intensity in aggressive responses [11]. According to Holldobler and Wilson, the significance of labor division determining each caste's specific duty at the colony level through synchronized cooperation among all individuals is a key factor in the colony's sustained survival [5]. Beyond age or sub-caste morphological differences, it will be necessary to determine the precise contribution of these novel, specialized catalyst individuals to the group's colony-level benefits [61]. Another essential element of the colony management system may be the exclusivity and unconventionality of these strategic individuals, who are physically distinct from the rest and who perform task allocation with creativity in contrast to the automatism-repetitive acts of their respective caste [61]. Observed sub-caste foraging (collective or solitary) and scouting activity in the field by these new workers suggests a degree of autonomy in task allocation decision-making. More observation is required to determine the precise contribution of this sub-caste to the colony's overall benefits.

In Indonesian oil palm plantations, the yellow larvae tended by *Oecophylla* workers within brood nests have been identified as *Liphyra brassolis*, an obligate myrmecophilous butterfly larva [62,63].

## 6. Conclusion

This study provided a comprehensive evaluation of the social structure composition of *O. smaragdina* colonies, revealing a population similar to that reported in other studies, except for the worker castes, comprised of extremely polymorphic individuals. The high fecundity of queens is supported by the abundance of eggs laid by thousands in brood nests, which is the first report of its kind for the *O. smaragdina* species. The population size of reproductive individuals can reach more than a dozen per nest and hundreds per colony. It is the first report to detect the existence of five distinct worker sub-castes (major big, major intermediate, intermediate 3 and 2, and minor worker) distinguished by size differences based on the head width, head, thorax, abdomen, and body lengths. The abdomen and body mass being the predictors of this significant difference are visually idiosyncratic (distinctive). The study's findings contrast with the conventional concept of the bimodal size frequency distribution model for worker ants by demonstrating the existence of a clear multimodal distribution. The study shows the difficulty of adult male collection due to their short lifespan and nest segregation. The study results indicated that environmental factors (rainfall and relative humidity) played a crucial role in the emergence of reproductive individuals. The precise function of the intermediate workers' contribution to the colony's survival requires

additional evaluation. The findings of this study demonstrate that the biological system of the *O. smaragdina* species serves as an investigative model for more diverse biological systems.

## Supporting information

**S1 Table. Colony eggs production.**
(DOCX)

**S2 Table. Shapiro-Wilk normality test – Tukey HSD.**
(DOCX)

**S3 Table. Box-Cox Transformation.**
(DOCX)

**S4 Table. Descriptive statistics. Sub-castes.**
(DOCX)

**S5 Table. Correlation Comparison by Ant Worker Sub-Caste – Spearman Correlation versus Pearson Correlation.**
(DOCX)

**S6 Table. Test of Multicollinearity.**
(DOCX)

**S7 Table. Descriptive analysis of *O. smaragdina* Felda I colony.**
(DOCX)

**S8 Table. Descriptive analysis of *O. smaragdina* matured big size colony UM III.**
(DOCX)

## Acknowledgments

This study is a part of the first author's PhD dissertation chapter. Special gratitude to the Malaysian Palm Oil Board (MPOB) and Felda Malaysia (Gunung Besout Sungkai Perak) for granting access to their facilities and plantations (providing support field staff for sampling). Thanks to Dr. Zubaidah Ya'cob from the Higher Centre of Excellence (HICoE) Tropical Infectious Diseases Research & Education Centre (TIDREC), Universiti Malaya and Dr. Salmah Yaakop, and Dr. Nurul Wahida Othman from Centre for Insect Systematics, Department of Biological Science and Biotechnology, Faculty of Science and Technology, Universiti Kebangsaan Malaysia, for providing all the investigative types of equipment for ant measurements, and photo recording. Special thanks to Associate Professor Dr. Roslinazairimah Zakaria, Centre for Mathematical Sciences, Universiti Malaysia Pahang Al-Sultan Abdullah, for providing the valuable R Core Team software to conduct all dataset analyses. Special thanks to the management of "Surau An-Nur, section 20 Shah Alam Selangor State" for providing accommodation and daily services helping to sustain the needs during the writing and correction period of the manuscript. EMP received a doctoral scholarship from the local government authority of the Martinique Island "Collectivité Territoriale de la Martinique-CTM.", under a French-European Union scheme, financial support (overseas travel, accommodation, daily life cost – not project funding) for students originating from the Caribbean Islands.

## Author contributions

**Conceptualization:** Moïse Pierre Exélis.

**Data curation:** Moïse Pierre Exélis.

**Formal analysis:** Moïse Pierre Exélis, Roslinazairimah Zakaria, Rabha W. Ibrahim.

**Funding acquisition:** Moïse Pierre Exélis.

**Investigation:** Moïse Pierre Exélis, Azarae Hj Idris, Zubaidah Ya'cob, Nurul Wahida Othman.

**Methodology:** Moïse Pierre Exélis, Roslinazairimah Zakaria.

**Project administration:** Moïse Pierre Exélis.

**Resources:** Moïse Pierre Exélis, Rosli Ramli, Azarae Hj Idris, Zubaidah Ya'cob, Salmah Yaakop, Nurul Wahida Othman.

**Software:** Roslinazairimah Zakaria, Rabha W. Ibrahim.

**Supervision:** Rosli Ramli, Azarae Hj Idris.

**Validation:** Moïse Pierre Exélis.

**Visualization:** Moïse Pierre Exélis, Roslinazairimah Zakaria, Rabha W. Ibrahim.

**Writing – original draft:** Moïse Pierre Exélis, Azarae Hj Idris.

**Writing – review & editing:** Moïse Pierre Exélis, Rosli Ramli, Roslinazairimah Zakaria, Zubaidah Ya'cob, Rabha W. Ibrahim, Salmah Yaakop, Nurul Wahida Othman.

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
