## [Decision Letter · Decision Letter 0]

4 Feb 2025

PONE-D-24-57771The social organization of the Asian weaver ant colonies: A natural enemy novel sub-castes worker’s functional activity findingsPLOS ONE

Dear Dr. Exélis,

Thank you for submitting your manuscript to PLOS ONE. After careful consideration, we feel that it has merit but does not fully meet PLOS ONE’s publication criteria as it currently stands. Therefore, we invite you to submit a revised version of the manuscript that addresses the points raised during the review process.

We look forward to receiving your revised manuscript.

Kind regards,

Munir Ahmad, PhD

Academic Editor

PLOS ONE

Journal Requirements:

2. Please include a copy of Table 4 which you refer to in your text on page 16.

Additional Editor Comments :

Based on both review report, there looks major concern about the duplication of findings with the pre-print as highlighted by the reviewer. Guidelines for improvement must be followed before submission of revised manuscript.

Statistical analysis needs major revision and must based on the parameters as suggested by the reviewers.

Figures unclear must be rechecked in the revised manuscript.

All suggestions point-wise must be answered properly.

Reviewers' comments:

Reviewer's Responses to Questions

**Comments to the Author**

1. Is the manuscript technically sound, and do the data support the conclusions?

Reviewer #1: Partly

Reviewer #2: Yes

2. Has the statistical analysis been performed appropriately and rigorously? 

Reviewer #1: No

Reviewer #2: Yes

3. Have the authors made all data underlying the findings in their manuscript fully available?

Reviewer #1: No

Reviewer #2: Yes

4. Is the manuscript presented in an intelligible fashion and written in standard English?

Reviewer #1: No

Reviewer #2: Yes

5. Review Comments to the Author

Reviewer #1: In the abstract: summary should be provided, without writing anything not known by the readers e.g. significant correlation - first report ?? Rewrite the abstract, without any unnecessary information.

I found the following publication on researchgate.net, which has almost similar content but posted to "preprints".

Please remove the same content from this manuscript and re-write this paper including the findings "NEW" ONLY here. Although the titles are different, many sections in both manuscripts overlap.

The social organisation of Oecophylla smaragdina (Hymenoptera:

Formicidae) colonies and their functional activity

Moïse Pierre Exélis1, 2, Azarae Hj Idris1, Zubaidah Ya’cob3, Rabha W. Ibrahim4, 5, 6, Salmah Yaakop7,

Nurul Wahida Othman7, Rosli Ramli1*

Statistical analysis:

Pearson's correlation should be applied to normally distributed data. your data normally distributed? Did you check?

Fig. 9: Kruskal Wallis mean plot - Is this a mean plot? Kruskal Wallis is about the median.

About intermediate worker castes mentioned in the manuscript:

Are you sure these are not "callow" major and minor workers? After the emergence from respective pupae, worker appearance including the size is different from that of mature workers (in most of the ant colonies including O. smaragdina, I observed this; also there are publications on other ant species.)

Reviewer #2: This manuscript is a very well-written paper.

But, tt is unclear whether the author accidentally omitted some figures or if there was a technical issue, but some figures are not included in the manuscript, such as fig. 14 A and B.

In the Fig. 5 D, male drone larva should be indicated with black arrow not circle.

The Fig 11 D is broken.

In line 406, please conduct a normality test and present the test statistic.

In line 481, please present the test statistics.

In line 520, I can't see the Fig. 5G.

In line 565, please present the test statistics.

In line 588, If there is a strong correlation between variables AL and BL, couldn't we consider these two variables to have collinearity?

In line 593, please present the test statistics.

In line 723, there is not referernce of Hasegwa (2013).

In line 724, There is not Table 4.

6. PLOS authors have the option to publish the peer review history of their article (what does this mean? ). If published, this will include your full peer review and any attached files.

**Do you want your identity to be public for this peer review?** For information about this choice, including consent withdrawal, please see our Privacy Policy .

Reviewer #1: No

Reviewer #2: No

---

## [Author Response · Author response to Decision Letter 1]

10 May 2025

The document is attached in this submission. Thank you.

---

## [Editor Report · Decision Letter 1]

24 May 2025

The social organization of the Asian weaver ant colonies: A natural enemy novel sub-castes worker’s functional activity findings

PONE-D-24-57771R1

Dear Dr. Exélis,

We’re pleased to inform you that your manuscript has been judged scientifically suitable for publication and will be formally accepted for publication once it meets all outstanding technical requirements.

Kind regards,

Munir Ahmad, PhD

Academic Editor

PLOS ONE
---

## [Editor Report · Acceptance letter]

PONE-D-24-57771R1

PLOS ONE

Dear Dr. Exélis,

I'm pleased to inform you that your manuscript has been deemed suitable for publication in PLOS ONE. Congratulations! Your manuscript is now being handed over to our production team.

Kind regards,

on behalf of

Dr. Munir Ahmad

Academic Editor

PLOS ONE